# Cellular features of localized microenvironments in human meniscal degeneration: a single-cell transcriptomic study

Weili Fu[1]*[†], Sijie Chen[2†], Runze Yang[1], Chen Li[2], Haoxiang Gao[2], Jian Li[1], Xuegong Zhang[2,3]*

[1]Department of Orthopedics, Orthopedic Research Institute, West China Hospital, Sichuan University, Chengdu, China; [2]MOE Key Lab of Bioinformatics, Bioinformatics Division, BNRIST and Department of Automation, Tsinghua University, Beijing, China; [3]School of Life Sciences and School of Medicine, Center for Synthetic and Systems Biology, Tsinghua University, Beijing, China

*For correspondence:
foxwin2008@163.com (WF);
zhangxg@tsinghua.edu.cn (XZ)

[†]These authors contributed equally to this work

Competing interest: The authors declare that no competing interests exist.

## Abstract

**Background:** Musculoskeletal tissue degeneration impairs the life quality and function of many people. Meniscus degeneration is a major origin of knee osteoarthritis and a common threat to athletic ability, but its cellular mechanism remains elusive.

**Methods:** We built a cell atlas of 12 healthy or degenerated human meniscus samples from the inner and outer meniscal zones of 8 patients using scRNA-seq to investigate meniscal microenvironment homeostasis and its changes in the degeneration process and verified findings with immunofluorescent imaging.

**Results:** We identified and localized cell types in inner and outer meniscus and found new chondrocyte subtypes associated with degeneration. The observations suggested understandings on how cellular compositions, functions, and interactions participated in degeneration, and on the possible loop-like interactions among extracellular matrix disassembly, angiogenesis, and inflammation in driving the degeneration.

**Conclusions:** The study provided a rich resource reflecting variations in the meniscal microenvironment during degeneration and suggested new cell subtypes as potential therapeutic targets. The hypothesized mechanism could also be a general model for other joint degenerations.

**Funding:** The National Natural Science Foundation of China (81972123, 82172508, 62050178, 61721003), the National Key Research and Development Program of China (2021YFF1200901), Fundamental Research Funds for the Central Universities (2015SCU04A40); The Innovative Spark Project of Sichuan University (2018SCUH0034); Sichuan Science and Technology Program (2020YFH0075); Chengdu Science and Technology Bureau Project (2019-YF05-00090-SN); 1.3.5 Project for Disciplines of Excellence of West China Hospital Sichuan University (ZYJC21030, ZY2017301); 1.3.5 Project for Disciplines of Excellence – Clinical Research Incubation Project, West China Hospital, Sichuan University (2019HXFH039).

## Editor's evaluation

This paper will be of interest to researchers studying meniscus homeostasis and knee osteoarthritis. It uncovers distinct subtypes of cell populations in the inner and outer parts of the human meniscus using single-cell RNA sequencing. In particular, this work further identifies how alterations

in meniscal cell populations may contribute to inflammation and osteoarthritis and thus serves as a resource paper for the field.

## Introduction

The knee is the largest and most complicated hinge joint in the human body. The meniscus is an important component of the knee joint with essential roles in the load-bearing, shock absorption, nutrition, and lubrication of the knee joint articular cartilage and is very susceptible to injury and has a limited reparative potential (*Murphy et al., 2019*; *Newman et al., 1989*; *Rai and McNulty, 2017*). The meniscus degeneration is a slow-developing disease of middle-aged or older people. Horizontal cleavage of the meniscus usually happens in the disease, and the lesion can be identified through knee MRI. Patients typically have no clear history of acute knee injury (*Beaufils et al., 2017*). Meniscal degeneration is one important risk factor for osteoarthritis (OA) and joint dysfunction, causing huge social and economic burdens (*Englund et al., 2012*). It has been found through clinical observations that nearly half of patients with meniscal degeneration will eventually develop OA over time (*Lohmander et al., 2007*). Studies have shown that inflammatory responses and biomineralization affect the meniscus cellular microenvironment and contribute to disease occurrence (*Sun et al., 2010*; *Goldring and Goldring, 2016*). However, the changes in the cell microenvironment in degenerated meniscus remain largely unclear. An adequate understanding of the changes in the degenerative meniscus is critical for preventing meniscal injury in young and middle-aged patients and for relieving symptoms in elderly patients with knee OA.

The meniscus can be anatomically divided into the inner region (the white–white zone) and the outer region (the red–red zone). Some researchers also use the 'red–white' zone to denote the in-between transition region. The inner and outer menisci differ in neurovascular distribution, structural composition, and recovery ability from injury. There are blood vessels and nerves in the outer area, while few blood vessels and nerves are observed in the inner part. The meniscal extracellular matrix (ECM) is the physical foundation of menisci's biological roles, and alterations in the ECM may lead to meniscus degeneration and dysfunction. Collagens and proteoglycan (PG) are ECM's two most important components. The outer meniscus is dominated by type I collagen, while the inner meniscus is dominated by type II collagen. The inner part of the meniscus has a relatively higher percentage of PGs compared with the outer. Aging and mechanical injuries may lead to meniscus degeneration. Different anatomical regions have varied recovery capabilities: the degeneration in the outer meniscus is more likely to heal, while the degeneration in the inner part tends to be irreversible (*Makris et al., 2011*; *Danzig et al., 1983*). The cellular and molecular basis behind anatomical regions remains to be explored. It is unclear what cell type contributes to the inner and outer variations and what cell types mediate the degeneration.

Building a detailed meniscal cell landscape is essential to understanding meniscus characteristics. Early profiling of meniscal cell heterogeneities dates back to the work of *Ghadially et al., 1983*. They studied injured and uninjured human menisci with the electron microscope and stated that chondrocytes, a few fibroblasts, myofibroblasts, and intermediate state cells between chondrocytes and fibroblast existed in menisci. Scotti et al. also discussed that the cells in the superficial zone were fusiform, while cells laid deeper were polygonal (*Scotti et al., 2013*). These classifications were mainly based on the shape and the surrounding matrix content. The recent development of single-cell omics has facilitated researchers' understanding of various cell types across multiple cartilage tissues, including knee joint cartilage (*Ji et al., 2019*), intervertebral disc (*Gan et al., 2021*), meniscus (*Sun et al., 2020*), etc. These researches initiated a rudiment of the chondrocyte cell types and states, covering parts of the heterogeneous cell subpopulations in the meniscus. *Sun et al., 2020* sketched the outline of the meniscal cell populations following the cell type definition convention of a knee articular cartilage study (*Ji et al., 2019*). They reported cartilage progenitor cells (CPC), regulatory chondrocytes (RegC), prehypertrophic chondrocytes (PreHTC), hypertrophic chondrocytes (HTC), fibrochondrocytes (FC), fibrochondrocytes progenitors (FCP), proliferating fibrochondrocytes (ProFC), degenerated progenitor (DegP), endothelial cells, and other immune cells in the meniscus, and showed links between meniscus progenitors and the progression of meniscal degeneration (*Sun et al., 2020*).

We aimed to build a refined atlas complementary to the existing data by systematically revealing cell heterogeneities in inner and outer meniscal zones under different health states. We created a

single-cell transcriptomic atlas of 45,744 cells from 12 healthy/degenerated meniscus samples of 8 patients and provided an online cell browser at http://meni.singlecell.info:3000/ to support flexible explorations of the data. We performed multiple rounds of cell-level quality control steps to remove and adjust noisy data such as empty droplets, doublets, dissociation-affected cells, and ambient RNA-polluted cells. Among the current single-cell sequencing studies involving articular cartilage and meniscus, we have a larger number of sequencing samples and cells. Our dataset is the first single-cell sequencing study data characterizing menisci's inner/outer zonation differences. We built a hierarchical cell type classification framework and identified five chondrocyte subtypes and two pericyte-like cell subtypes. Functional enrichment analyses suggested these subtypes may have specialized ECM construction and remodeling duties. We profiled various immune cell types and observed damage-induced inflammatory responses in degenerated samples. Taken together, we inferred a hypothetical model of ECM disassembly, inflammation, and angiogenesis in the meniscal tissues. The model may also fit degenerative situations in other cartilage tissues, such as articular cartilage and intervertebral disc, because they contain similar cell types and ECM contents (*Chen et al., 2017b*). Our discoveries may suggest new clues for treating various types of joint degeneration in the human body.

## Methods

### Human meniscus cell sample preparation

Four degenerative meniscus specimens were obtained from patients with severe OA who underwent total knee arthroplasty, and four normal meniscus specimens were obtained from patients with bone tumors or severe trauma who underwent amputation. The meniscus tissues were removed from the patient's knee joints and divided into the inner (white–white zone) and outer (red–red zone) areas. The degeneration group is comprised of patients A, B, C, and D; and the normal group is comprised of patients E, F, G, and H. Patients A, B, E, and F contributed paired inner–outer samples. Patients C and H contributed unpaired inner samples, and patients D and G contributed unpaired outer samples because the other pairs had obtained in the surgery had poor sample states. Detailed designs of experiments can be found in *Supplementary file 1*.

After carefully dividing the meniscus tissue into the inner and outer parts, the menisci were cut into small pieces. We used 0.25% Trypsin (Gibco 25200072) to digest the pieces for 0.5 hr under 37°C. Next, we used 2 mg/ml collagenase IV to dissociate the pieces for 4–6 hr under 37°C and finally obtained dissociated cells. The dissociated cells were resuspended at a concentration of 250–1200 cells/μl and with viability between 67–87% for microfluidics (Chromium Single-Cell Controller, 10× Genomics). According to the manufacturer's instructions, cells were loaded into the chip and run using the Chromium Single Cell 3′ Reagent Kit v2 (10× Genomics). In brief, Gel beads with GemCode barcodes and primers and individual cells were encapsulated in oil droplets. Next, within each oil droplet, mRNA were released from lysed cells, barcoded, reverse transcribed to cDNA, and sequenced. Sequencing-ready single-cell transcriptome libraries were mapped to the human reference genome GRCh38-3.0.0 and quantified by CellRanger 3.1.0.

### Multiplex immunofluorescence (OPAL) staining

Human meniscus normal and degenerated samples were fixed in 4% paraformaldehyde and embedded in paraffin. We sliced the embedded paraffin samples in series, and each slice was 4 μm thick. Firstly, water-bath heating was used for antigen retrieval. Then various cell marker primary antibodies were incubated with the paraffin slide of meniscus tissue to conduct the continuous staining with the Opal Polaris Multiple-Color Manual IHC Kit (NEL861001KT). We used different primary antibodies to simultaneously label Ch.1 (SERPINA1), Ch.2 (MMP14), Ch.4 (CYP1B1), and PCL (ACTA2) on the same tissue slide. The automated staining system (BOND-RX, Leica Microsystems, Vista, CA) was used to perform the chromogen-based multiplex immunohistochemistry labeling. SERPINA1 (ab207303), MMP14 (ab51074), CYP1B1 (ab33586), and ACTA2 (ab5694) were all purchased from Abcam and diluted at a concentration of 1:100. The Opal Polaris dyes were used to pair with these antibodies containing fluorophores for tyramide signal amplification to enhance sensitivity. The sequence of labeling for detecting each marker was optimized: CYP1B1 (Opal 570), MMP14 (Opal 620), SERPINA1 (Opal 690), ACTA2 (Opal 780), and DAPI (4′,6-diamidino-2-phenylindole). Including an autofluorescence section, the staining process is the same as above, but no primary antibody is added. Multiplex analysis was

operated to analyze the results of simultaneously stained tissue slides. In addition, common immuno-fluorescence staining was performed using CDON (ab227056), CD31 (ab9498), and CD45 (ab40763) labeled Ch.3, endothelial cells, and immune cells. All experiments have three biological replicates.

## Single-cell data quality control steps

We adjusted the ambient RNA expression with SoupX (*Young and Behjati, 2020*) v1.2.1 and generated read counts free of background noises for all downstream analyses. We used scanpy (*Wolf et al., 2018*) and Seurat (*Stuart et al., 2019*; *Butler et al., 2018*) to conduct the basic filtering out outliers cells with high n_genes metrics (the number of observed genes per cell), high n_counts metrics (the number of UMI per cell), and high percent_mito metrics (the mitochondrial gene fraction). Next, we assigned draft cell type identities to the cells using SingleR (*Aran et al., 2019*). Note that the SingleR-generated cell type labels were only used to perform quality control analysis to avoid removing biologically meaningful cells with poor data qualities. We calculated doublet scores using Doublet-Finder (*McGinnis et al., 2019*) removed the cells with high doublet scores and outlier clusters with high doublet scores. We observed that DoubletFinder tends to identify small clusters as doublets incorrectly, so we assert these small populations as singlets if they had well-defined identities given by SingleR or appear clear biological functions such as cell cycles. To remove the confounding factor introduced by the dissociation-induced gene expression (*van den Brink et al., 2017*; *Denisenko et al., 2020*), we decided to remove the heavily affected cells by the dissociation steps. We calculated dissociation scores using methods described in *van den Brink et al., 2017*.

Raw single-cell RNA-seq datasets contain numerous low-quality droplets. Hence, we should ensure the majority of the data items we analyzed correspond to viable cells. It is known that one simple quality control filtering on the aforementioned metrics (n_genes, n_counts, percent_mt) can't remove all of them. We then conducted comprehensive quality control steps that can be summarized as (1) top-down refinements and (2) divide and conquer. First, we divide cells into several rough populations with distinct biological identities, for example, chondrocytes, immune cells, endothelial cells, etc. We zoomed into these populations one by one and performed subclustering within them. The within-population subclustering often works like a centrifuge – the low-quality cells mixed in the large rough population usually form tiny outlier groups when subclustering is performed on a single population. When a tiny group is mainly comprised of cells with high doublet scores, cells with high mitochondrial gene fractions, cells with high dissociation scores, we consider removing it. After manually inspecting and dropping these tiny outlier groups, we finally get much cleaner datasets.

## Single-cell data clustering steps

To eliminate the genetic background variations and technical noises of the single-cell RNA-seq data, we used Harmony (*Korsunsky et al., 2019*) to perform integrative clustering across different samples. Major clusters corresponded to chondral cells, ACTA2$^+$ cells, endothelial cells, and immune cells were identified across samples and validated with cluster correlation analysis (*Figure 1—figure supplement 1C* and *Figure 2—figure supplement 1A*).

To understand large cell populations with higher resolutions, we subset the chondral and *ACTA2$^+$* majors clusters, performed reclustering on harmony-derived embeddings, and obtained multiple fine clusters. To alleviate the scRNA-seq data noise and promote data consistency across samples (*Hua and Zhang, 2019*), we partitioned original cell data points into homogenous meta-cells with MetaCell (*Baran et al., 2019*). We then used HGC (*Zou et al., 2021*) to build hierarchical relationships based on the meta-cells.

## Differentially expressed genes analysis

Differentially expressed genes (DEGs) were identified by the 'FindMarkers' function and the 'FindAllMarkers' function provided by the Seurat package. The DEGs of each chondrocyte cluster were identified by comparing the cluster with all the other clusters. A full list of DEGs in shown in Figure 2 is given in *Supplementary file 2*. The health group (degeneration/normal) markers were calculated in an overall comparison way and a per-cluster comparison way. We first extracted the total chondrocytes (from Ch.1 to Ch.5) and compared them between the degenerated and normal groups. We made the same comparison to the total PCL populations. The health group markers are reported in *Figure 3—figure supplement 5A*.

## Gene set enrichment analysis and gene set variation analysis

Enrichment analyses were performed with top fold-change DEGs and a BH-adjusted p-value threshold of 0.05, using R package clusterProfiler (*Wu et al., 2021*). Cluster-wise enriched terms (Ch.1–5 and PCL.1/2, *Figure 2—figure supplement 2*) were obtained using cluster marker DEGs (one cluster vs. the others). Health state-specific terms were obtained using status DEGs (all chondrocytes in normal samples vs. all those in the degenerated) and displayed in *Figure 3—figure supplement 5B, C*.

We assigned gene set activity scores to individual cells using gene set variation analysis (GSVA) (*Hänzelmann et al., 2013*), with gene sets obtained from MSigDB (*Subramanian et al., 2005*; *Liberzon et al., 2015*). The GSVA-derived gene set scores were visualized in Figures 2G and 3C–F. The GSVA analyses used the following MSigDB-curated gene ontology terms: GO0050900 (leukocyte migration), GO1905563 (negative regulation of vascular endothelial cell proliferation), GO1904018 (positive regulation of vasculature development), GO0071772 (response to BMP), GO0030199 (collagen fibril organization), GO0070555 (response to interleukin 1), GO0071559 (response to transforming growth factor beta), GO:0034612 (response to tumor necrosis factor), and GO0030239 (myofibril assembly).

## Gene regulation network inference

We used the python package pySCENIC (*Van de Sande et al., 2020*; *Aibar et al., 2017*) to infer gene regulatory networks (GRN) from single-cell expressional profiles. The workflow includes three substeps: identifying coexpression modules, trimming modules with prior knowledge, and calculating activating scores.

The workflow builds coexpression modules by choosing TFs that predict target gene expression well. First, the algorithm builds gradient boosting machine models predicting each target gene's expression from TF expressions and keeps the models' feature weights as a measure of TF–target regulatory scores. Second, the algorithm creates modules by adding/filtering weighted links between TFs and their targets. To reduce the stochasticity of the results, the algorithm independently uses six weight filtering rules to build modules. The six rules are: (1) keep TF–target pairs with weights > the 75% percentile of the weights; (2) keep TF–target pairs with weights > the 90% percentile of the weights; (3) for each TF, keep the top 50 TF–target pairs with the highest weights; (4) for each target, keep top 5 TF–target pairs with highest weights; (5) for each target, keep top 10 TF–target pairs with highest weights; (6) for each target, keep top 50 TF–target pairs with highest weights. After this step, many TF–target modules are created. The algorithm only keeps TF–target links that have positive regulation relationships (the Pearson correlation of TF and target expressions >0.03) and drops modules with <20 genes inside (TF itself included). Finally, we get a list of primary coexpression modules, each containing a TF and its positively regulated targets.

The module trimming process combines the regulatory motif information with the coexpression analyses. Given the coexpression-derived modules, we then keep target genes that are overlapped with TF's regulatory motif region. We adopt the 10 kb up-/downstream regions of a target gene's TSS as the gene's distal enhancer occurrence region and the 500 bp upstream/100 bp downstream region as the gene's proximal promoter occurrence region. After trimming, more reliable sets of modules are generated.

Finally, we quantified and binarized the activation levels of the TF–target modules with AUCell (*Aibar et al., 2017*) and visualized them with ComplexHeatmap (*Gu et al., 2016*) and cytoscape (*Shannon et al., 2003*).

## Cell crosstalk analysis

Given the averaged normalized expression value of a ligand gene in a sender cluster $i$ as $E_{L,i}$, and that of a receptor gene in receiver cluster $j$ as $E_{R,j}$, we defined the crosstalk between cluster $i$ and cluster $j$ via the ligand–receptor pair L–R is their product $E_{L,i}E_{R,j}$. We enumerated all ligand and receptor genes provided by CellTalkDB (*Shao et al., 2021*) and calculated crosstalk products for sender–receiver cluster pairs. We empirically evaluated the statistical significance of this product by counting the more extreme null products after shuffling the cluster labels. The overall crosstalk level between the sender cluster $i$ and the receiver cluster $j$ is defined as $\sum_L \sum_R E_{L,i}E_{R,j}$ if $E_{L,i}E_{R,j} > \theta$, and $\theta = 1.5$ in our cases.

We compared the crosstalk variations between the degenerated and the normal group by subtracting the products in two conditions (visualized *Figure 5—figure supplement 2*). We identified

ligand–receptor interactions that were high in the normal/degenerated group and grouped the differential interactions into ECM, TNF, TGFβ (Transforming growth factor-β), chemokine, and cytokine antigen-presentation categories (*Figure 5—figure supplement 2A–L*).

### Online cell browser construction

We used the python package cellxgene to provide online cell browser service. The matrices along with other metadata were saved in scanpy h5ad format for cellxgene.

## Results

### The cellular landscape of the inner and outer parts of human menisci

To understand the composition and molecular profiles of the cells from different regions and degeneration states, we sampled four healthy menisci and four degenerated menisci from a patient cohort and performed single-cell RNA sequencing. *Figure 1A* illustrates the morphological patterns of the samples: Meniscal specimens in the normal group are smooth and complete with a clear structure. Specimens from the degenerated group are swollen, irregular in shape, damaged in structure, and vascularized. Each specimen's inner part (white–white zone) and the outer part (red–red zone) were strictly separated and collected. These samples are dissociated and prepared for single-cell RNA sequencing (Methods), as shown in the overall workflow (*Figure 1B*).

After comprehensive quality control steps removing low-quality cells and inferred doublets (Methods), we recovered 45,744 cells from 12 samples of 8 patients (*Supplementary file 1*). The number of counts observed per cell and the number of genes observed per cell are visualized in *Figure 1—figure supplement 1A, B*. We can infer from the quality control steps that the general sequencing quality is good except for the E2 outer and H12 inner samples.

We conducted a survey on the inner and outer menisci's major cell types. An unsupervised clustering algorithm partitioned the cells into separated clusters (Methods), which were visualized using uniform manifold approximation and projection (UMAP). UMAP and stacked bar plots in *Figure 1C–E* described the distributions of cells in the inner menisci, and *Figure 1F–H* described them in the outer menisci. *Figure 1C, F* visualized the origin of the cells, indicating that the batch effects of the samples were well eliminated, and the biological variations were preserved. The cell type correlation plots in *Figure 1—figure supplement 1C* also confirmed that cells of the same type, not the same batch, had higher affinities. We observed five major cell types in the inner menisci: chondrocytes, lymphocytes, myeloid cells, endothelial cells, and a group of *ACTA2*⁺ cells. We also observed the five types in the outer menisci, plus a group of Schwann cells, indicating there are nerve tissues in the outer menisci. The percentages of cells in *Figure 1E, H* showed that, in general, the outer menisci had higher cell type diversities. In the more diverse outer menisci, the degenerated group (A1, B2, and D2) had higher immune cell ratios and lower *ACTA2*⁺ cell ratios.

The cell types were determined by combinations of markers in *Figure 1I, J* and *Figure 1—figure supplement 2A, B*. We identified cell clusters as chondrocytes if they expressed a high level of cartilage-related genes such as *DCN*, *PRG4*, *COL1A2*, *COL3A1*, *ACAN*, *COMP*, *FN1*, *CHI3L1*, and *CHI3L2*. We identified cell clusters with high expression levels of *PLVAP*, *PECAM1* as endothelial cells. Immune cells were identified using high *PTPRC*, myeloid markers *CD68*, *C1QA*, *HLA-DQA1*, and T cell markers *CD3D*, *CD3E*. In addition, there are a group of cells separated from other cells in both inner and outer areas of the meniscus, which has high expression levels of *ACTA2*. Besides high expressions of *ACTA2*, this group of cells also expressed some chondrocyte-associated gene markers, such as *COL1A2*, *COL3A1*, *DCN*, and *FN1*. We tentatively named this *ACTA2*⁺ group 'pericyte-like cells' or PCL (*Figure 1I, J* and *Figure 1—figure supplement 2A, B*) and reanalyzed them together with the chondrocytes in *Figure 2* and associated texts.

### Functional coordination of the heterogeneous chondrocytes and PCL cells

We built a comprehensive cell identity framework by charting the meniscal chondrocytes and PCL cells into a hierarchical coordination system. Unlike the other single-cell cartilage studies that assigned cell types with several marker genes, we defined cell types in a top-down way to simultaneously reveal subtle granular-view variations and exhibit general similarities.

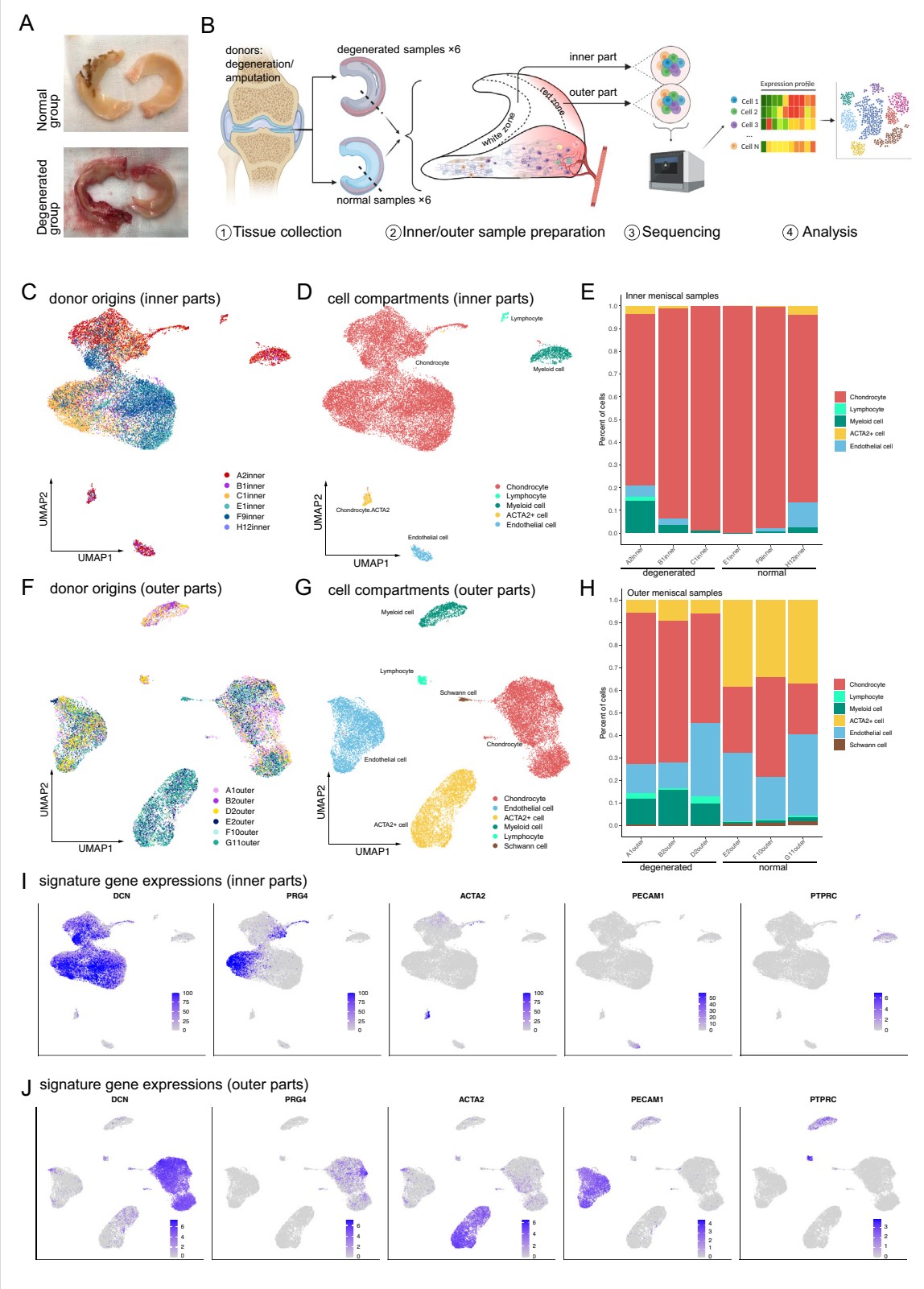

**Figure 1.** Single-cell RNA-seq reveals major cell classes in human menisci. (**A**) Photographs of typical normal and degenerative meniscus specimens. (**B**) The overall workflow of the single-cell sequencing. Inner and outer meniscal parts are collected separately from patients with normal/degenerated menisci. (**C, D**) Distributions of cells in the inner meniscus samples. Cells of the degeneration batches and the normal batches are integrated together. (**C**) Uniform manifold approximation and projection (UMAP) visualization of donor origins (inner parts). (**D**) UMAP visualization of the cell identity

*Figure 1 continued on next page*

*Figure 1 continued*

compartments (inner parts). (**E**) The percentages of the identified cell classes in six samples of the inner meniscus. (**F, G**) Distributions of cells in the outer meniscus samples. Cells of the degeneration batches and the normal batches were integrated together. UMAP visualization of the donor origins and cell identity compartments (outer parts). (**H**) The percentages of the classes in six samples of the outer meniscus. (**I, J**) The expression levels of the major class markers of the inner cells (upper) and the outer cells (lower). Darker colors indicate higher expression levels.

The online version of this article includes the following figure supplement(s) for figure 1:

**Figure supplement 1.** Quality control metrics for single-cell RNA sequencing.

**Figure supplement 2.** Signature genes of the large cell classes.

To develop such a cell identity framework, we took multiple factors into account: clustering labels, clustering granularities, marker gene expression specificity, generalization abilities across samples and conditions, supporting evidences in other studies, matrix compositions, and biological functions (Methods). We finally identified five chondral clusters named chondrocyte-1 to chondrocyte-5 (Ch.1–Ch.5 in *Figure 2A*) and two pericyte-like cell clusters (PCL.1 and PCL.2 in *Figure 2A*) named ACTA2$^+$ cells in *Figure 1D, G*. The distribution shown in *Figure 2B, C* shows cells from different batches were mixed well after data integration. The correlations illustrated in *Figure 2—figure supplement 1A* and the marker gene heatmap in *Figure 2—figure supplement 1B* indicated that inner and outer samples had consistent expression profiles. From these analyses, we can infer that batch effects are resolved well, and biological variations were retained.

We visualized the cell type identification framework in a tree in *Figure 2D*, which assigned labels to cells with hierarchical marker combinations. Common fibroblast markers like *DCN* and *LUM* are highly expressed in the chondral class. Smooth muscle cell and pericyte markers like *ACTA2*, *TAGLN*, and *MYL9* are highly expressed in the PCL class. More detailed marker expressions in chondrocyte and PCL subpopulations were visualized in a heatmap *Figure 2F* and *Figure 2—figure supplement 1C*. The marker genes differentially expressed in each subpopulation shown in the heatmap are given in *Supplementary file 2*.

For more granular cell subtypes, we analyzed their biological functions using GSVA (*Hänzelmann et al., 2013*) in *Figure 3C–F*, *Figure 2—figure supplement 2H* and gene set overrepresentation enrichment analysis (GSEA) (*Wu et al., 2021*) in *Figure 2—figure supplement 2A–G*. The gene expressions and function predictions suggested subtypes dominating homeostasis states, subtypes associated with the pro-/anti-angiogenesis process, as well as subtypes associated with the construction/disassembly of the ECM.

We constructed GRN using pySCENIC (*Aibar et al., 2017*; *Van de Sande et al., 2020*) to decipher the common and unique programs behind the chondrocyte and the PCL populations. In the inferred GRN, links originate from the upstream TFs to the downstream targets. Each TF forms a module of nodes that contains its targets. We calculated the module activation scores and showed the binarized states of the gene regulatory programs in *Figure 2—figure supplement 3A*. *Figure 2—figure supplement 3B* visualizes a core part of TFs and targets in the GRN. Several polygons were added to annotate subcluster-specific GRN modules. We found that PCLs shared common gene regulatory modules represented by *STAT4*, *NR2F2*, and chondrocyte subcluster Ch 2–5 shared *TRPS1*, *FOXC1*, *HOXD10*, and each subcluster possessed some specific modules. For example, Ch.2 had *KLF6*, *FOXN3*, *AKR1A1*, Ch.3 had *SIX3*, Ch.4 had *CEBPA*, and Ch.5 had *MYBL2*. This GRN analysis also explained intra-PCL heterogeneities and indicated PCL cells were a distinct population with divergent gene regulatory programs, although they have some collagen-producing transcriptomic features like chondrocytes.

## The characteristics and function predictions of PCL cells in the meniscal microenvironment

The PCL cells present strong muscle contractile gene expression signatures (*Figure 2F*), including *ACTA2*, *MYL9*, *TAGLN*, etc. *ACTA2*, which encodes a smooth muscle actin involved in vascular contractility, as well as *TAGLN* (transgelin) and *MYH/MYL* (myosin heavy/light chains) genes. These genes have been used for identifying pericytes or smooth muscle cells across multiple human organs, including intervertebral disc (*Gan et al., 2021*), intestinal tract (*Elmentaite et al., 2021*), heart (*Litvinukova et al., 2020*), brain (*Smyth et al., 2018*), etc. The gene set enrichment analyses of these

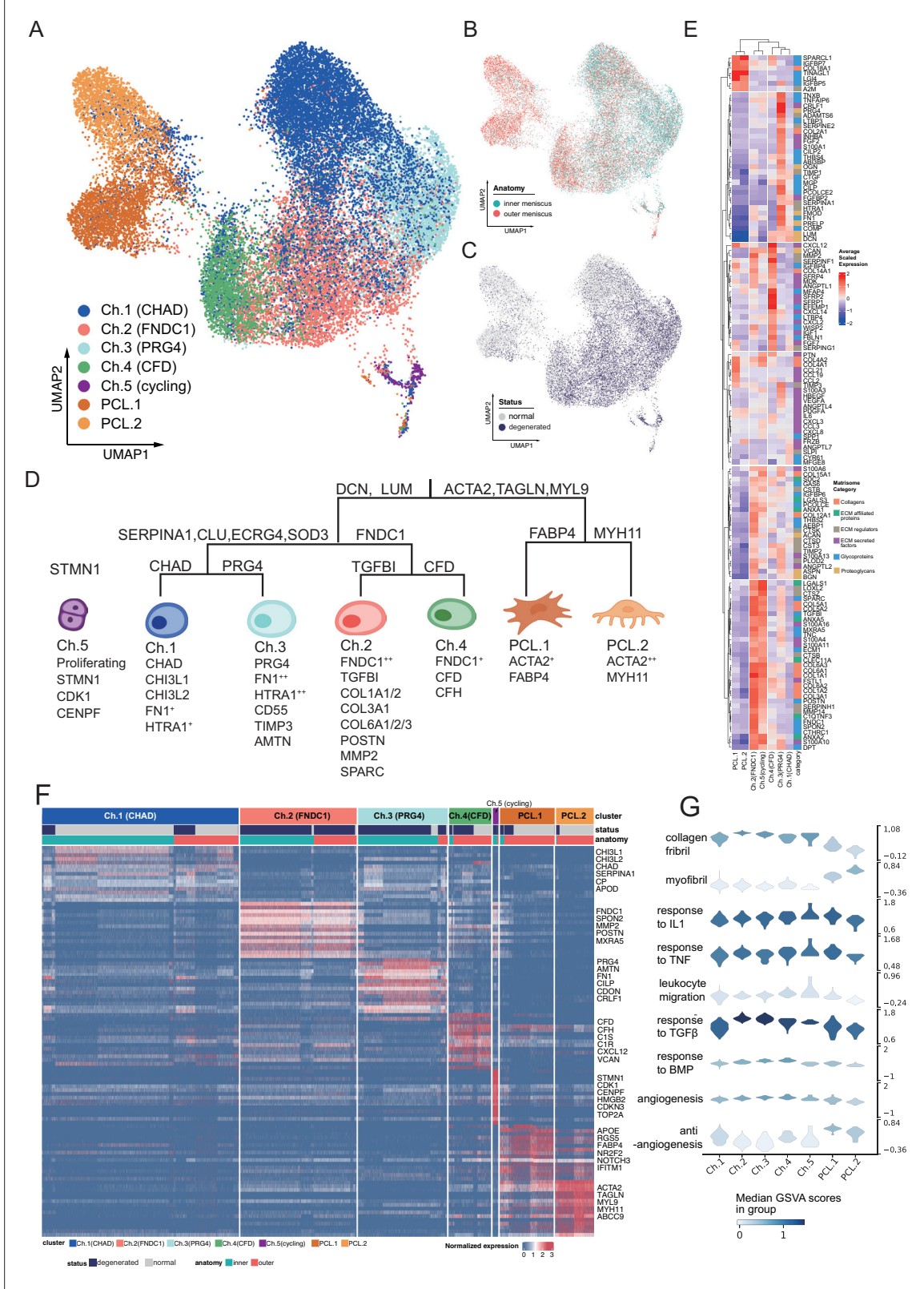

**Figure 2.** Identification of chondrocyte and PCL subclusters in human meniscus. (**A**) Uniform manifold approximation and projection (UMAP) visualization of the chondrocyte and PCL class cells. (**B, C**) UMAP visualization of the distribution of chondrocytes at different anatomical sites (up) and different sample statuses (down). (**D**) A schematic diagram of the hierarchical classification of chondrocyte subgroups. The classification criteria are along the tree path, and each group's highly expressed marker genes are given below each cluster. (**E**) A cluster-level heatmap shows extracellular matrix

*Figure 2 continued on next page*

*Figure 2 continued*

(ECM)-related gene expressions in each chondrocyte/PCL subpopulation. (**F**) A cell-level heatmap reveals the normalized expression of differentially expressed genes for each cluster defined in (**D**). (**G**) Violin plots score nine molecular themes of each cell type (leukocyte migration, suppression of angiogenesis, angiogenesis, response to BMP, collagen fibril, response to IL1, response to TGFβ, response to TNF, and myofibril). We calculated the scores using gene set variation analysis (GSVA) with gene sets picked from MSigDB.

The online version of this article includes the following figure supplement(s) for figure 2:

**Figure supplement 1.** Label agreements of the chondrocyte and pericyte-like cells across samples.

**Figure supplement 2.** Gene ontology enrichment of chondrocyte and PCL subcluster markers.

**Figure supplement 3.** Regulons derived by pySCENIC.

**Figure supplement 4.** Public chondrocyte datasets reanalysis (I).

signature genes suggested the PCL population has a discrepant molecular identity from the chondrocyte lineage because GO terms such as 'muscle contraction', 'muscle organ development', and 'smooth muscle cell proliferation' were enriched (*Figure 2—figure supplement 2F*).

In the work of *Sun et al., 2020*, they named a similar *ACTA2*[+] *MYLK*[+] *MCAM*[+] *MYL9*[+]cluster as the FCP and claimed FCP could differentiate into degenerated meniscus progenitors (DegP) in degenerated menisci, or into adipogenic/osteogenic lineages when cultured ex vivo. We did not name the population we found as FCP for three reasons: Firstly, enrichment analyses showed its similarity to pericytes/smooth muscle cells. Secondly, our PCL population showed weak differentiation potential into DegP because this group of cells in our data did not show concentrated expressions of DegP markers *GREM1*, *CDCP1*, and *DNER*. Thirdly, the only expressions of *GREM1*, *CDCP1*, and *DNER* were sporadically distributed on the UMAP plot (*Figure 2—figure supplement 4B*) and topologically separated from our *ACTA2*[+] *MYLK*[+] *MCAM*[+] *MYL9*[+]PCL cluster (*Figure 2A*, *Figure 2—figure supplement 4A*).

We named this population as PCL considering multiple factors. The PCL cells are more abundant in the vascularized outer meniscus region than in the avascular inner part (*Figure 1E, H*). Our immunofluorescence imaging found that this population of cells lay around the vascular endothelial cells and formed a tube shape outside the blood vessels (*Figure 4A, B*). Like the meniscus, the intervertebral disc also has cartilage tissue and blood vessel structures. *Gan et al., 2021* named a group of *ACTA2*[+] *MCAM*[+] *TAGLN*[+] in the human intervertebral disc as pericytes, which can be a reference for defining meniscal cell clusters. For these reasons, we inferred that the *ACTA2*[+] *MYL9*[+] *TAGLN* [+] population in our meniscus samples belong to the mural structure around the meniscal vasculature and chose to name this cluster as the 'pericyte-like cell' considering its signature gene expressions, functional analyses, and spatial distributions. We admit that it is difficult to fully distinguish pericyte from smooth muscle cells purely based on the transcriptomes because the two cell types share numerous markers, and their markers appear to be dynamic and affected by the tissue types and pathogenic states (*van Dijk et al., 2015*).

It is known that pericytes attach to the surface of vascular endothelial cells and maintain vasculature stability via the endothelium–pericyte crosstalk (*Eilken et al., 2017*; *Sugihara et al., 2020*). We therefore inferred that these cells in the meniscal microenvironments could maintain vascular stability like pericytes according to the GSVA results and the decreased percentages of PCLs in the degeneration group. Previous studies have shown that pericytes have pro-angiogenesis capabilities (*Eilken et al., 2017*; *Armulik et al., 2011*). In our datasets, we observed expressions of multiple potential angiogenesis inhibition genes. For example, in *Figure 3—figure supplement 1*, we showed that both PCL.1 and PCL.2 expressed ECM components Collagen IVα1 (*COL4A1*), Collagen IVα2 (*COL4A2*), and Collagen XVIII (*COL18A1*), the non-collagenous fragments of which (i.e., arresten, canstatin, and endostatin, respectively) were known to have anti-angiogenesis functions (*de Castro Brás and Frangiannis, 2020*; *Marneros and Olsen, 2001*). We also plotted multiple other pro-/anti-angiogenesis factors' expressions in *Figure 3—figure supplement 1B* to have a full view of the pro-/anti-effects and found the pro-angiogenesis genes were seldom expressed in PCL either.

We identified two subtle subclusters within the PCL class. One subcluster PCL.1 expresses a higher level of *FABP4*, *RGS5*, and the other PCL.2 subcluster has relatively higher expressions of *MYH11*, *MYH9*, *CNN1*, and *TAGLN*. *Kumar et al., 2017* analyzed these genes' functions in the specification and diversification of pericyte/smooth muscle cells from mesenchymoangioblasts. The GSVA suggested

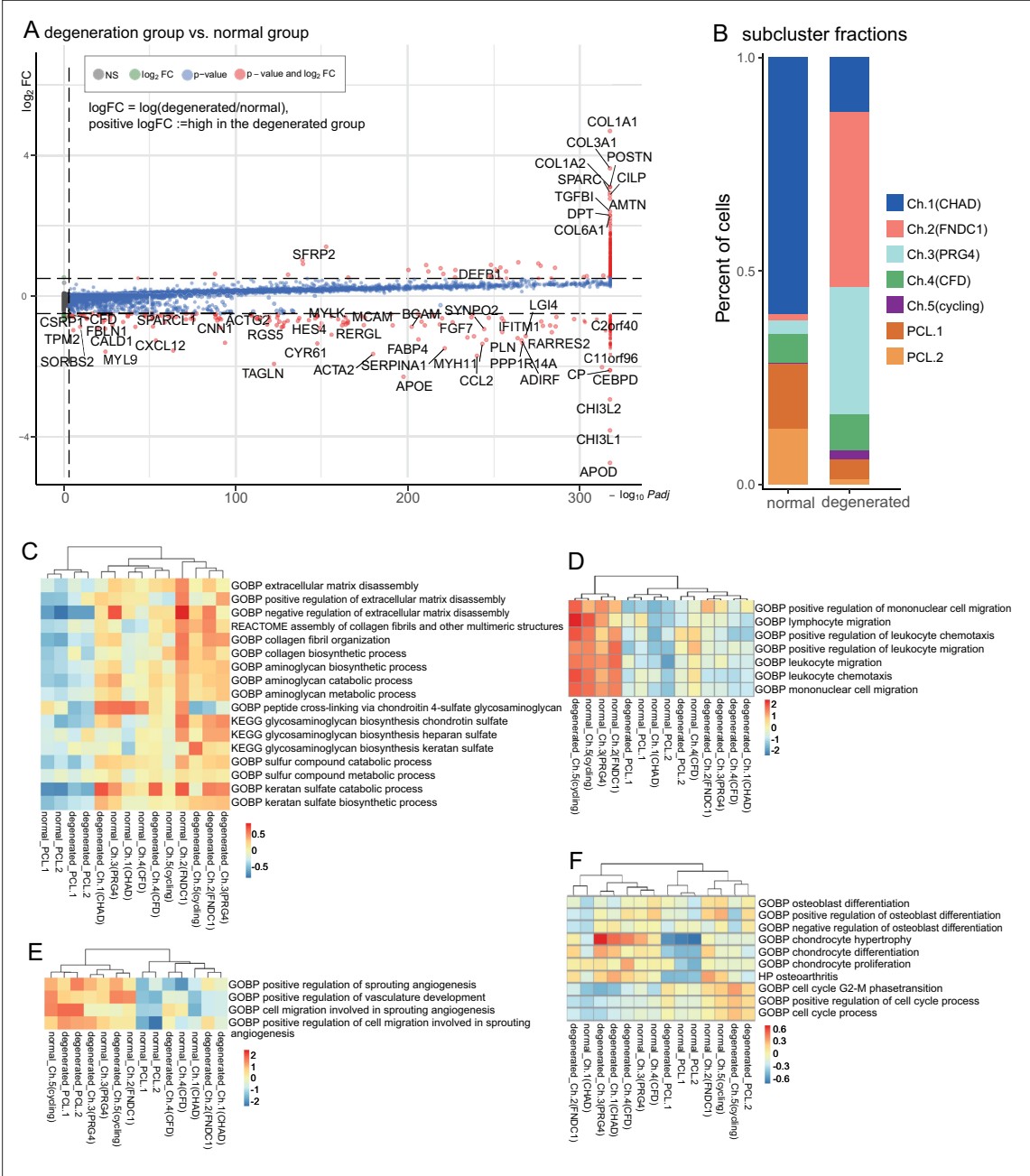

**Figure 3.** Degeneration molecular patterns in chondrocytes and PCLs. (**A**) A volcano plot shows differentially expressed genes with high fold-change values. The comparison was made between (1) degenerated chondrocytes/PCL cells and (2) normal chondrocytes/PCL cells. (**B**) The proportion of each subcluster of chondrocytes in the degenerated and normal meniscus. (**C–F**) Gene set variation analysis (GSVA) of each cluster under degenerated/normal conditions. Gene sets supported functions are evaluated using GSVA (Methods), and scaled values are visualized in the heatmaps. We observed Ch.2 and Ch.3 had high scores for matrix disassembly activities, angiogenesis activities, chemotaxis activities, and chondrocyte hypertrophy regardless of the normal/degeneration status. The cycling chondrocytes Ch.5 have high scores in the cell-cycle-related terms.

The online version of this article includes the following figure supplement(s) for figure 3:

**Figure supplement 1.** Expression levels of selected gene sets.

**Figure supplement 2.** Public chondrocyte datasets reanalysis (II).

**Figure supplement 3.** Comprehensive integration of public data and in-house data.

**Figure supplement 4.** Finding differentially expressed genes (DEGs) in total chondrocytes and PCLs (degenerated vs. normal).

**Figure supplement 5.** Interpreting differentially expressed genes (DEGs) in total chondrocytes and PCLs (degenerated vs. normal).

*Figure 3 continued on next page*

*Figure 3 continued*

**Figure supplement 6.** Finding composition changes and differentially expressed genes (DEGs) in individual chondrocyte and PCL clusters (degenerated vs. normal).

**Figure supplement 7.** Shifts of cell type compositions and inflammation states.

that PCL.1 had more active responses to the pro-inflammatory cytokines IL-1 and TNF-α (*Figure 2G*) and expressed higher levels of chemokines genes *CCL2*, *CCL19*, *CCL21*, *CXCL12* (*Figure 3—figure supplement 1D*), and *COL4A1/2* (*Figure 3—figure supplement 1E*). The GSVA scores suggested that PCL.2 had higher myofibril gene set activation scores, indicating this population could have higher muscle contractility and may regulate the blood flow in the microvasculature (*van Dijk et al., 2015*).

## Chondrocyte subpopulations and their potential roles in ECM remodeling

The ECMs in meniscal tissues undergo continuous remodeling processes to keep homeostasis of the microenvironment. We aimed to reveal meniscal chondrocyte heterogeneities and to find their associations with the ECM homeostasis and remodeling in normal and degeneration states. With the hierarchical cell type definition framework, we identified two major branches and a small cluster undergoing cell-cycle phase transitions in the chondrocyte class. These branches of cells were predicted to have varied roles in the construction and maintenance of the ECM, angiogenesis, and leukocyte chemotaxis.

The largest branch is the chondrocyte-1 (Ch.1) population that highly expresses *CHAD*, *CHI3L1*, *CHI3L2*, *ECRG* (*C2orf40*), and *APOD*. Some of these genes (e.g., *CHI3L1/2*) have been described as the markers of RegC (*Ji et al., 2019*). This cluster's percentage decreased in the degeneration group. The gene set activation analysis indicated that this population of cells had anti-angiogenesis effects (*Figure 2G*). In *Figure 2E*, we visualized the gene expressions that play essential roles in the ECM according to the Matrisome project (*Naba et al., 2012*). ECM-related gene expressions indicated Ch.1 highly expressed ECM regulators *TIMP1* and *TIMP4*, which were inhibitors of ECM destruction proteins. Hence, we inferred that the CHAD⁺ Ch.1 population is a cluster of chondrocytes that maintain homeostasis in the meniscal tissue. We named another part of this branch that expresses high levels of *PRG4*, *HTRA1*, *FN1*, *CD55*, *TIMP3*, *CDON*, *AMTN* as Ch.3 (PRG4). We inferred that Ch.3 cells had lubrication function because the protein proteoglycan 4 (*PRG4*) has a lubrification function on the articular cartilage surface, and these cells were located near the surface of the meniscus. Our immunofluorescence staining further confirmed that its marker CDON expresses along the surface region of the meniscus (*Figure 4*). This population highly expresses *TIMP1/2/3* (*Figure 3—figure supplement 1A*) and may inhibit the matrix decomposition by opposing the *MMP/ADAM/ADAMTS* family enzymes.

One branch that highly expresses *FNDC1* and *THY1* has two subclusters. One *TGFBI*⁺ subcluster in this branch has particularly high expression levels of *FNDC1*, so we named it chondrocyte2-FNDC1 (Ch.2). The Ch.2 population highly expresses OA marker genes (*Fisch et al., 2018*) and chondrocyte hypertrophy genes (*Ji et al., 2019*) such as type I, III, and VI collagen genes (*COL1A1/2*, *COL3A1*, *COL6A1/2/3*), *POSTN*, *MMP2*, *SPARC*, and *MXRA5* (*Figure 3—figure supplement 2A, B*). Among these upregulated genes, *FNDC1* and *MXRA5* have been recently reported as signatures of chondrocyte-mediated valve calcification in aortic stenosis (*Bouchareb et al., 2021*). It is also known that *MXRA5* (matrix remodeling-associated protein 5) is a TGF-β1 regulated gene (*Poveda et al., 2017*), which is consistent with the high GSVA scores of 'Response to TGFβ' in Ch.2 shown in *Figure 2G*. Besides *MMP2*, this population also expresses other *MMP*, *ADAM*, and *ADAMTS* family genes, such as *MMP11/13/14*, *ADAM9/12*, and *ADAMTS2*, which can target collagen, aggrecans or other ECM components (*Figure 3—figure supplement 1A*). The increased expression of these genes indicated that Ch.2 (*FNDC1*) is a chondrocyte population associated with aberrant ECM degradation and remodeling, and this population may contribute to the progress of human meniscal degeneration. The other *CFD*⁺ subcluster in this branch expresses high levels of *CFD*, *CFH*, *C1R*, *C1S*, *CXCL12*, and *VCAN*. We name it Ch.4 (*CFD*). This subcluster expresses genes associated with alternative complement activation pathways (*C1R*, *C1S*, *C2*, *C3*, *C6*, and *C7*) and may promote inflammation by recruiting macrophages and neutrophils (*Figure 3—figure supplement 1C*).

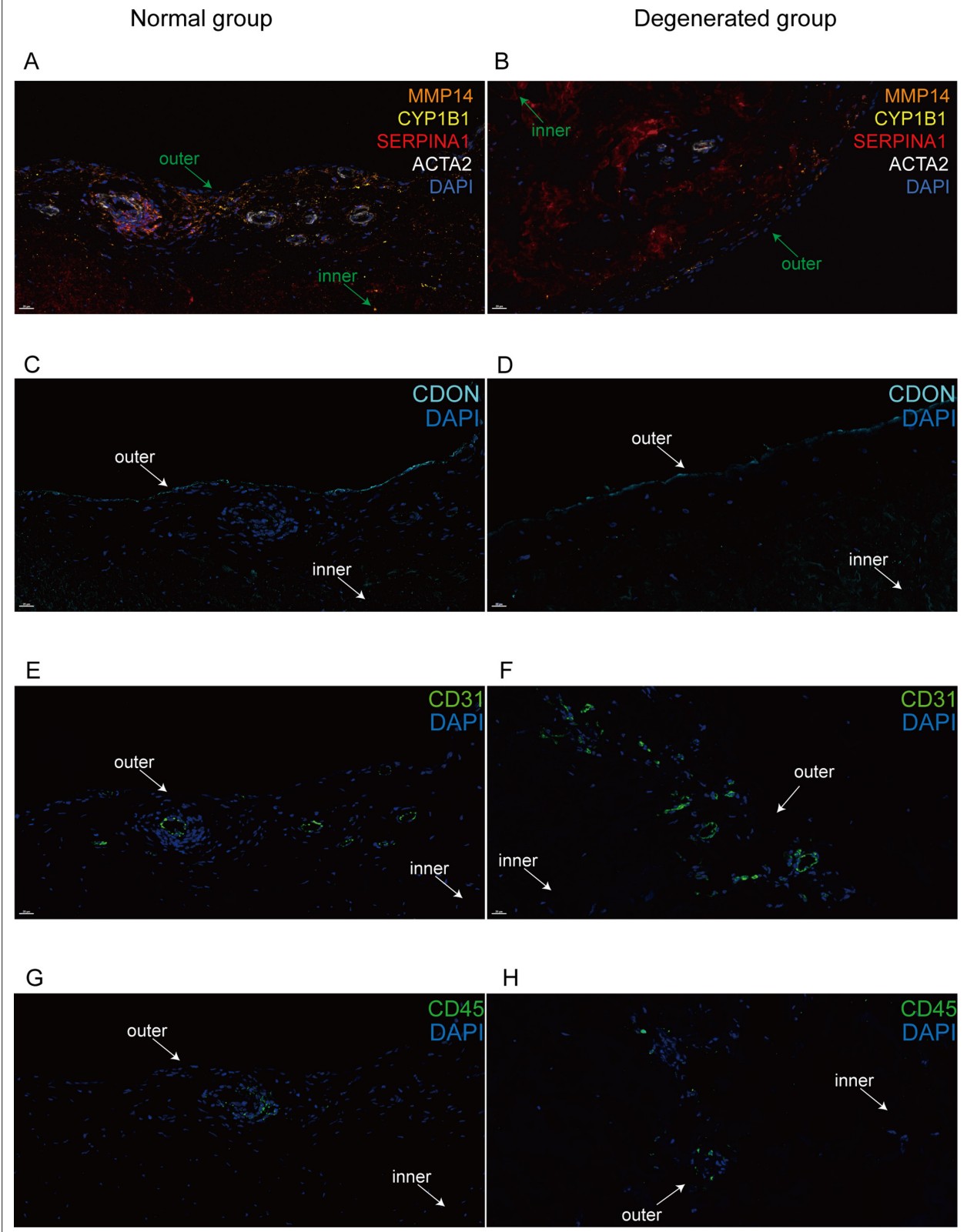

**Figure 4.** Immunofluorescent staining of human menisci demonstrating resident cell subtypes. Representative immunofluorescence staining images. Color settings: MMP14 for Ch.2 cells (orange), SERPINA1 for Ch.1 cells (magenta), ACTA2 for PCL cells (white), CDON for Ch.3 cells (cyan), CYP1B1 for Ch.4 cells (yellow), CD31 for endothelial cells (green), and CD45 for immune cells (green). Nuclei are stained blue (DAPI). Scale bar 20 μm. (**A**) Multiplexed IF staining of the normal meniscus. (**B**) Multiplex IF staining of the degenerated meniscus. (**C**) IF staining of CDON in a normal sample.

*Figure 4 continued on next page*

*Figure 4 continued*

(**D**) IF staining of CDON in a degenerated sample. (**E**) IF staining of CD31 in a normal sample. (**F**) IF staining of CD31 in a degenerated sample. (**G**) IF staining of CD45 in a normal sample. (**H**) IF staining of CD45 in a degenerated sample (IF: immunofluorescence). The inner and outer zones of menisci were annotated with arrows.

Ch.5 represents chondrocytes in proliferation states. They highly expressed the S- and G2M-phase cell-cycle genes such as *STMN1*, *CDK1*, *CENPF*, *HMGB2*, *CDKN3*, *TOP2A* and were mostly found in the degenerated samples. It is generally thought that cells during mitosis condense their chromatins, downregulated genes that maintain cell identities, and reestablish the gene expression patterns when the mitosis exits (*Palozola et al., 2017*). Hence, it is hard to judge whether Ch.5 is a high-stemness chondrocyte progenitor cluster or a dividing differentiated cell cluster based on these highly expressed markers. We may assume that the cells, though in a proliferation state, retained some transcripts before mitosis, and we may infer their origins or fates via the reference mapping analysis. We re-assigned Ch.5 cell identities using cells in Ch.1–Ch.4 clusters as a reference with SingleR (*Aran et al., 2019*) and found 98.63% Ch.5 cells were mapped to Ch.2 and the rest were mapped to Ch.3 (*Figure 3—figure supplement 7F*). This identity mapping suggested the proliferative Ch.5 could be the origin of the Ch.2 cells increased in the degenerated group.

We did not directly adopt the chondrocyte terminology used in other studies such as *Ji et al., 2019* and *Sun et al., 2020*, but instead used Ch.1–Ch.5 to represent the subclusters because our data spans chondrocytes of higher quantity and greater heterogeneity. We have examined the literature markers on our data in *Figure 2—figure supplement 4A, B*. Those markers did not show highly concentrated expressional patterns. We reanalyzed the data in *Sun et al., 2020* and validated our markers (*Figure 3—figure supplement 2C–E*). We found Ch.1 markers *CHI3L1/2* expressed on clusters 1, 2, 7, Ch.2 markers *FNDC1* expressed on clusters 0 and 3, and cycling cell markers *STMN1* expressed on clusters 5 and 6. The expressions of *FNDC1*, *TGFBI*, *PRG4*, *CFD*, and *CFH* were highly overlapped with each other, which is slightly different with our data (*Figure 3—figure supplement 2E*). We performed a comprehensive integration of our data and the public data in *Sun et al., 2020* using Seurat (*Stuart et al., 2019*). *Figure 3—figure supplement 3A, B* shows the transferred UMAP coordinates after integration. From the comprehensive integration results, we inferred that our data spanned broader transcriptomic states and the public data mainly corresponded to a subset near Ch.2, Ch.3, and Ch.5 locations. We calculated the percentages of transferred labels in the public clusters (*Figure 3—figure supplement 3C–E*) and revealed their mutual correspondences. We found the *ACTA2*+ cluster 3 in public data partially mapped to the PCL cells in our data. We used a hierarchically clustered joint heatmap of our data and public data to visualize the cluster correspondences (*Figure 3—figure supplement 3F*). Considering these integration results, we believe our cluster labels and markers span a larger state space and could better represent cellular heterogeneities.

## Degeneration and zonation-associated cell subpopulation changes

We compared the overall gene expression differences between the normal and degenerations conditions for chondral and PCL populations and derived a series of conditional DEGs. We identified genes upregulated in the degeneration group, such as *COL1A1*, *COL1A2*, *COL3A1*, *COL6A1*, *POSTN*, *SPARC*, *CILP*, *TGFBI*, *DPT*, and *ATMN*, and genes upregulated in the normal group, such as *APOD*, *CHI3L1*, *CHI3L2*, *CEBPD*, *C11orf96*, *C2orf40* (*ECRG4*), *ACTA2*, *TAGLN*, *BCAM*, and *MCAM* (shown in *Figure 3A* and *Figure 3—figure supplement 4*, an extended list of genes in *Figure 3—figure supplement 5A*). The enrichment analysis interpreting the potential functions of these DEGs is available in *Figure 3—figure supplement 5B, C*. We believe the overall expression differences are mainly contributed by the cell type compositional changes because these genes are identified as markers of the chondrocyte subpopulations. For example, the expressions of Ch.2 marker genes *COL1A1/2*, *COL3A1*, *COL6A1*, and *SPARC* are higher in the degenerated group; and Ch.3 markers *AMTN* and *CILP* also have higher expressions in the degenerated group; whereas PCL cell markers *ACTA2*, *TAGLN*, *BCAM*, and *MCAM* are highly expressed in the normal group. All these expression differences are consistent with our compositional observations that Ch.2 and Ch.3 increased in degeneration and Ch.1 and PCL decreased in degeneration (*Figure 3B* and *Figure 3—figure supplement 6A*). Besides overall differences, we also calculated the DEGs in each cluster between the degenerated and normal conditions (*Figure 3—figure supplement 6B*) and found more subtle expressional

shifts in the degeneration states. Considering all factors above, it appears that the overall expressional changes after degeneration could be explained by the increase of Ch.2 and Ch.3 as well as the decrease of PCL cells.

To explore each subcluster's biofunction in degeneration, we performed GSVA analyses of each population in two conditions. We first analyzed the GSVA scores of ECM-related gene ontology terms in *Figure 3C* and found Ch.2 could have high ECM disassembly scores while Ch.3 could negatively regulate the disassembly regardless of the degeneration states. They could play essential roles in the anabolic/catabolic balance. Ch.2 and Ch.3 also got high scores in leukocyte chemotaxis (*Figure 3D*), angiogenesis (*Figure 3E*), and chondrocyte hypertrophy-associated terms (*Figure 3F*); while in contrast, Ch.1 had relatively lower scores on these terms. This is consistent with the fact that Ch.1 cells dominate the normal meniscus (*Figure 3B*), and leukocyte infiltration and angiogenesis mainly happen in the degenerated meniscus. Ch.5 got high scores on the cell-cycle-related terms (*Figure 3F*). PCL cells got higher pro-angiogenesis scores in degenerated samples than in normal samples. The GSVA analyses indicated the increased or decreased subclusters in degenerated samples had pro- or anti-degeneration functions and further supported that the compositional changes of subclusters primarily contributed to the degeneration process.

It is acknowledged that the inner and outer zones have varied recovery abilities, ECM compositions, cellular compositions, and mechanical characteristics. But it is still unknown which zone is the primary contributor to joint inflammation in OA. We analyzed the zonation-dependent cellular state shift and computationally evaluated inflammation response scores in two regions. From the cell density plots in *Figure 3—figure supplement 7A, B*, we found that the inner zone chondrocytes shift from the Ch.1 positions to the Ch.3 positions, while the outer zone chondrocytes shift to Ch.2 and Ch.4 positions. Both zones have cell state shifts after degeneration. We compared the cell densities instead of directly comparing the subpopulation percentages. Since the molecular profiles of subclusters vary on a continuum and the clustering boundaries could be vague, directly comparing the percentages with paired/unpaired statistical tests could miss the state shifts (*Figure 3—figure supplement 7C, D*). Compared to Ch.3, the Ch.2 cells have higher scores on the responses to inflammation cytokines TNF and IL1 (*Figure 2—figure supplement 2H*). We also evaluated the GSVA scores of three inflammation gene sets on the chondrocytes and found the outer-degenerated groups consistently got significantly higher inflammation scores than the outer-normal group. In contrast, the inner zone cells did not have this trend (*Figure 3—figure supplement 7E*). From the computational analyses, we inferred that outer menisci could contribute more to joint inflammation in OA.

## Spatially resolve cell subtype distributions with in situ imaging

We used immunofluorescence staining to validate the subpopulation markers and study the spatial distribution of the subclusters in the human meniscus. Among the top 20 marker genes (according to log2 fold-change values) for each chondrocyte subpopulation, we chose SERPINA1, MMP14, CDON, CYP1B1, ACTA2, CD31, and CD45 for staining. Normal and degenerated meniscus sample slides were simultaneously stained with the following antibodies: anti-SERPINA1 for Ch1(*CHAD*), anti-MMP14 for Ch2(*FNDC1*), anti-CYPIB1 for Ch4(*CFD*), and anti-ACTA2 for PCL cells. We used anti-CD31, anti-CDON, and anti-CD45 to label vascular endothelial cells, Ch3(*PRG4*), and immune cells for immunofluorescence staining. From the staining results, we can clearly find that CDON + chondrocytes were mainly distributed on the surface of the meniscus tissue, and ACTA2[+] pericytes were presented surrounding blood vessels. SERPINA1[+], CYP1B1[+], and MMP14[+] chondrocytes were mixed and scattered inside the meniscus. Immune cells were mainly distributed around the blood vessels in the meniscus. Compared with the normal samples, the number of ACTA2[+] chondrocytes was decreased, the number of *MMP14*[+] chondrocytes was markedly increased, and the number of vascular endothelial cells was significantly increased in degenerative specimens (*Figure 4A–H*).

## Crosstalk changes of chondrocytes, leukocytes, and endothelial cells in meniscus degeneration

Besides chondrocytes, various types of leukocytes, and endothelial cells also have indispensable roles in the occurrence and development of cartilage degeneration and OA (*Hsueh et al., 2021*; *Alahdal et al., 2021*).

For the leukocyte part, we identified macrophages/monocytes, neutrophils, mast cells, dendritic cells, T cells, and cycling immune cells in normal and degenerated meniscal samples (*Figure 5A, B*, *Figure 5—figure supplement 1A*). Macrophage/monocyte makes up the largest portion of the total immune population. The percentages of macrophage/monocyte, T cell, dendritic cell, and total immune cell increased in the degeneration group, which was consistent with the vascularization and the damage-induced inflammation (*Figure 5B, C*). We noticed some pro-inflammatory cytokine/chemokine genes upregulated in degeneration samples, for example, interleukin genes *IL1A*, *IL1B*, *IL6*, *IL15*, and *IL18* upregulated in the cycling subpopulation (*Figure 5A*); chemokine genes *CXCL1*, *CCL3*, and *CXCL3* upregulated in macrophages/monocytes (*Figure 5—figure supplement 1D*). We also noticed increased expressions of metallothionein genes in the degenerated group, for example, *MT1X* and *MT1G* in macrophages/monocytes (*Figure 5—figure supplement 1B*), *MT1X*, *MT2A*, and *MT1E* in T cells (*Figure 5—figure supplement 1C*).

We identified an arterial–capillary–venous gradient in the meniscal endothelial cell population (*Figure 5E–G*) with other organs' endothelial markers (*Schupp et al., 2021*; *Kalucka et al., 2020*). The capillary–arterial endothelial cells highly express *GJA5*; the capillary endothelial cells highly expressed *PLVAP*; and the capillary–venous endothelial cells highly expressed *VWF*. These endothelial cells comprise the microvascular and capillary vessels. A tiny cluster of lymphatic endothelial cells was also observed, which may come from the lymphatic vessels in the meniscus. We compared the total endothelial cells between the normal/degenerated groups and reported a list of DEGs (*Figure 5H*). In the normal group, *CST3*, *ACKR1*, *RBP7*, and *IGFBP4* were upregulated, while in the degenerated group, *S100A4*, *DDIT4*, *CRIP1*, and *MGP* were upregulated. *CST3* is ranked first in the normal group DEGs, which was reported to have inhibition effects on the proliferation, migration, tube formation, and permeability of endothelial cells (*Li et al., 2018*). *ACKR1* is also highly expressed in the normal group, which encodes a chemokine decoy receptor inhibiting the effectiveness of other chemokines (*Rot, 2005*). These molecules contributed to the blood vessel's stability in the normal group.

With all major classes of meniscal cells being identified, we inferred the cell–cell interactions (CCIs) based on the ligand and receptor expressions and compared the interaction intensities between the normal/degenerated samples. The results suggested that CCI intensities generally increased in the degeneration group, especially for the endothelial cells and immune cells (*Figure 5I*). These enhanced CCIs are also consistent with the angiogenesis and leukocyte recruitment phenomena in the degeneration group (*Figure 6*). *Figure 5J* shows some instances of the significant CCIs originating from the sender cell type to the receiver cell type. For example, in the degenerated meniscus, we observed that a chemokine gene *CXCL8* was upregulated by macrophages/monocytes, neutrophils, DCs; and its receptor gene *SDC2* was expressed in chondrocytes and the pericyte-like cells. We also observed significant *CALM1/2-INSR* CCIs occurred in the degeneration in which angiogenesis happened – many cell types upregulated *CALM1/2*, and endothelial cells upregulated the insulin receptor gene *INSR*. This observation is consistent with a previous study showing insulin receptors' pro-angiogenesis functions. (*Walker et al., 2021*). We also identified multiple other cell–cell crosstalks enhanced in the degeneration group that are involved in immunoregulations. They are mediated by TNF superfamily interactions (*TNF*, *TNFSF10 → VSIR*, *TNFRSF1A/B*), and TGF superfamily interactions (*INHBA → ENG*), and Class I HLA → *APLP2* interactions. In *Figure 5—figure supplement 2A–L*, we provided a detailed resource of the predicted CCIs. We illustrated sender–receiver CCI pairs high in the normal or degeneration groups. Note that all the CCIs mentioned above were derived using gene expressions and curated lists of ligands and receptors. We did not examine these computational inferences with experiments.

Based on the above observations, we inferred that the immune cells and endothelial cells could be important regulators for chondrocytes and ECMs (*Figure 6*). It has been known that the degenerated ECM released damage-associated molecular patterns (DAMPs) that initiated inflammatory responses (*Martel-Pelletier et al., 2016*; *Chen et al., 2017a*). According to our inference, immune cells could sense the DAMP signals, migrate to the inflammatory meniscus and produce cytokines and chemokines. The inflammation could in turn enhance ECM catabolic enzymes like *MMP/ADAM/ADAMTS*, stimulate blood vessel proliferation, and further promote immune infiltration in the meniscal tissue. These factors work in a manner like a 'positive feedback loop'. In common conditions, the DAMP-induced inflammation will reverse when the DAMP molecules are cleared (*Zindel and Kubes, 2020*).

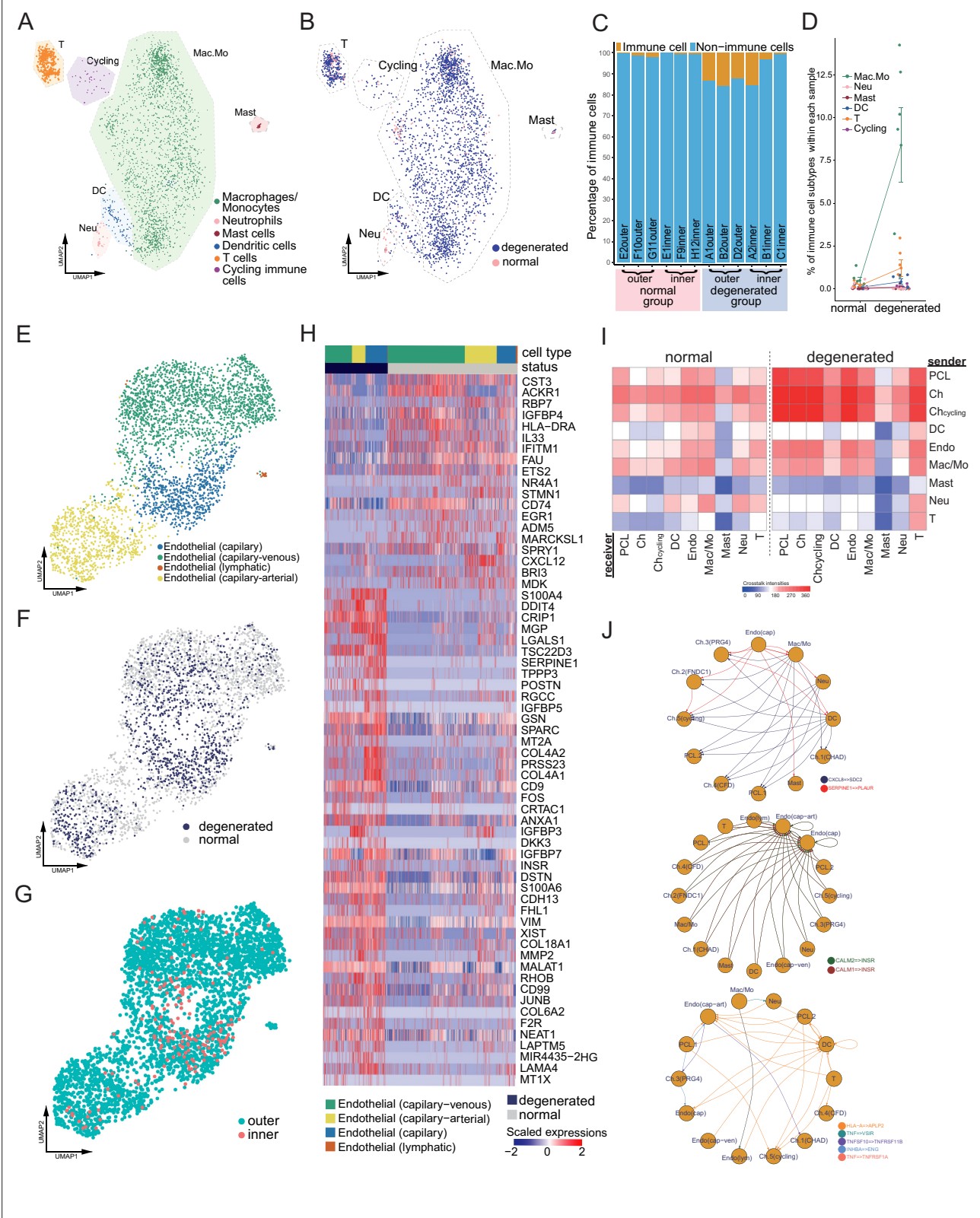

**Figure 5.** Immune, endothelial cells, and their crosstalk with chondrocytes. (**A**) Uniform manifold approximation and projection (UMAP) visualization of the immune cell types in the meniscus. (**B**) UMAP visualization of the statuses of samples (degenerated/normal). (**C**) Per-sample bar plots visualize the immune cell percentage changes between the normal and degenerated group. (**D**) Changes in immune cell percent between normal and degenerated groups. Error bars show the standard deviations of the data. Wilcoxon p values: Mac.Mo 0.004, Cycling 0.10, DC 0.14, Neu 0.19, T 0.22, Mast 1.0. (**E**)

*Figure 5 continued on next page*

*Figure 5 continued*

UMAP visualization of the endothelial class cells' subtypes. (**F**) UMAP visualization of sample status. (**G**) UMAP visualization of the anatomical regions. (**H**) Top differentially expressed genes between different health states (degenerated vs. normal). The heatmap shows *z*-score-scaled gene expression values. (**I**) General cell–cell crosstalk between large populations. The ligand–receptor pair crosstalk was evaluated at the large population level (Methods). (**J**) Representative crosstalk was significantly enhanced in the degeneration group.

The online version of this article includes the following figure supplement(s) for figure 5:

**Figure supplement 1.** Immune cell signature genes.

**Figure supplement 2.** Ligand–receptor pairs that were up-/downregulated in degeneration.

However, if the above factors work in the positive feedback loop in some perturbed conditions, they can mutually promote each other and drive the microenvironment away from homeostasis.

## Discussion

Recent single-cell studies have shown the importance of meniscal progenitors in tissue engineering and given a rough picture of the meniscal cell types. However, the existing data were still insufficient to reveal degeneration participants in the microenvironment. In this work, we investigated meniscal chondrocytes, developed a hierarchical classification system, and classified the cells into five major

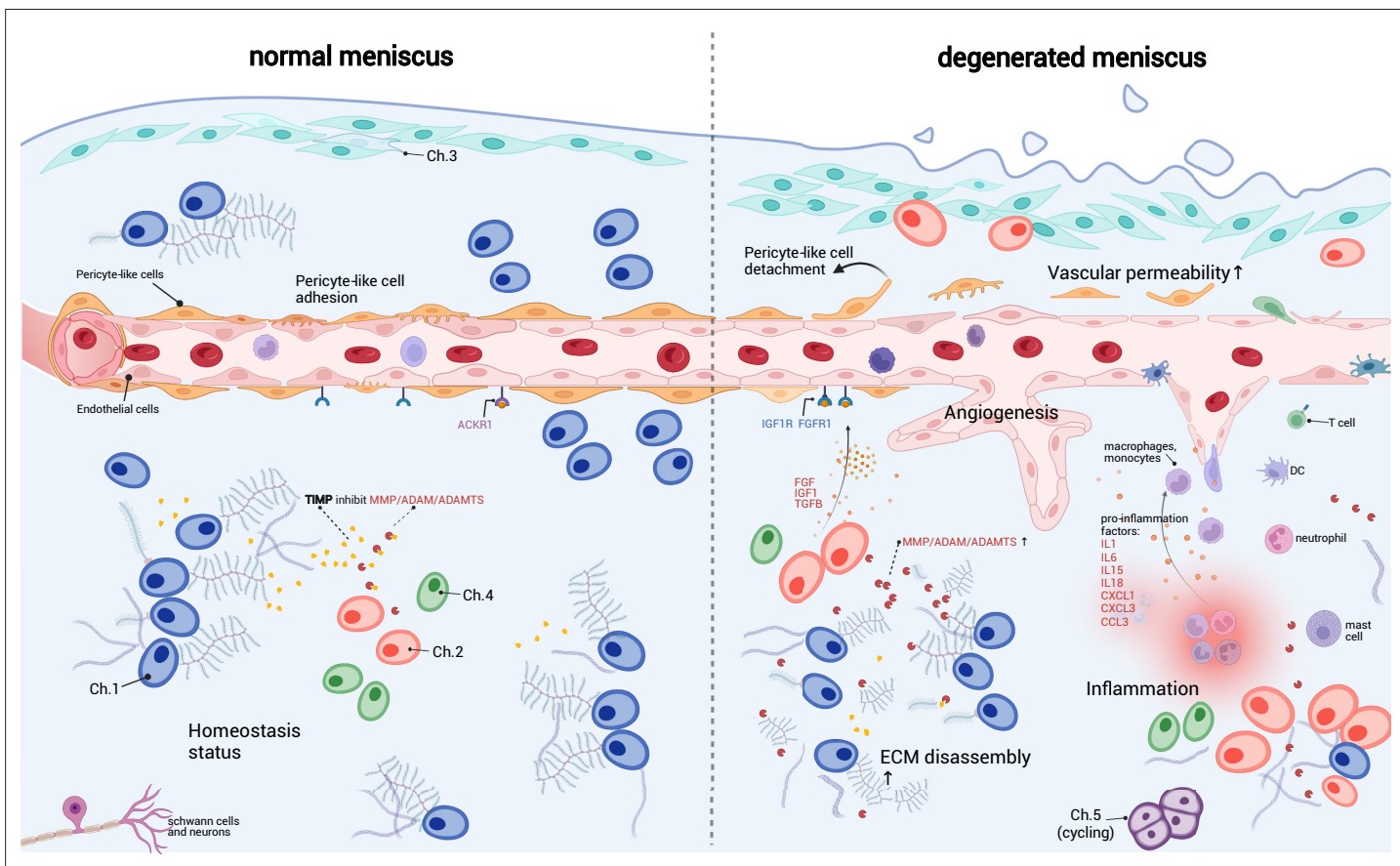

**Figure 6.** A schematic diagram of the microenvironment changes between the normal and the degenerated meniscus. The left side visualizes the homeostasis meniscus, where Ch.1 is the dominant chondrocyte population. In this situation, the extracellular matrix (ECM) decomposition and synthesis reach a dynamic equilibrium, and the aberrant proliferation of blood vessels is inhibited. The right side visualizes the degenerated meniscus, where the orchestrated microenvironment balance is broken. The pericyte-like cells detach from the blood vessels and lower the stability of the blood vessels. The endothelial cells grew and formed new blood vessels in the degenerated area. The vascular permeability increased and allowed more immune cells to infiltrate the degenerated tissue. Ch.2 and Ch.3 increased in quantity and produced more matrix disassembly enzymes, damaging the ECM, releasing angiogenesis factors, and recruiting more immune cells. The immune cells, such as macrophage/monocytes, DCs, and T cells, also have chemotaxis functions. There are also cycling chondrocytes and nerve tissue cells in the meniscus.

populations (Ch.1–Ch.5). We identified endothelial cells enriched in both outer meniscal parts and inner parts, which are typically conceived as avascular 'white–white zone'. We inferred that these unanticipated cells were the degeneration product because newly generated vascular endothelial cells usually occur in cartilage as OA progresses (*Goldring and Goldring, 2016*). Surrounding the microvessels are the pericyte-like cells possessing muscle contraction features. We also observed Schwann cells, which typically wrap around the neurons, indicating there are nerve tissues in the human meniscus. For the immune cells, we observed monocytes/macrophages, DCs, T cells, neutrophils, mast cells, and some cycling immune cells.

The meniscal microenvironment changes significantly from the homeostasis when the degeneration progresses (*Figure 6*). In normal meniscal tissues, the synthesis and disassembly of the matrix molecules were balanced in the orchestrated ECM metabolism. We inferred that the dominant Ch.1 population was essential to the balance maintenance because they expressed TIMP family molecules that inhibited the ECM decomposers (MMP/ADAM/ADAMTS). The PCL population that adhered to the surface of endothelial cells maintained vasculature stability. The high-level expressions of atypical chemokine receptors like *ACKR1* neutralized the chemokines and attenuated the angiogenesis (*Xu et al., 2007*). The patrolling immune cells mainly pass by the tissue in the blood vessel without residing or being attracted.

However, the balanced microenvironment was altered significantly in the degenerated meniscal tissues. The expression of ECM decomposers (MMP/ADAM/ADAMTS family molecules) from chondral subpopulations Ch.2 and Ch.3 might surpass the inhibition forces (TIMP) and decompose the ECM by breaking large PG molecules into fragments. We believe the blood vessels have two sides in the homeostasis of the meniscus. On the one hand, the blood vessels can help to restore acute injuries to health; on the other hand, the proliferated blood vessels can build pathways for the immune cells so that they can infiltrate the meniscal matrix and cause inflammation. We observed multiple synergistic factors promoting angiogenesis and increased endothelial percentages in the degenerated samples. While the pericyte stabilizes the blood vessel in health, the detachment and decrease of the pericyte may lead to pathological states (*Figueiredo et al., 2020*; *Armulik et al., 2011*; *Diéguez-Hurtado et al., 2019*). On the vascular wall, the endothelial cells could downregulate ACKR1 expressions, respond to the pro-angiogenesis factors like VEGFA, FGF, IGF1, TGFβ produced by the Ch.2, Ch.3 Ch.4, grew new branches, increased the vascular permeability. Changing the chondrocyte subtype ratios and limiting blood vessel proliferation may delay or reverse the meniscus degeneration.

In addition to these findings from the analyses of the meniscal single-cell sequencing data, we have also made an interactive website to store and let readers to browse the processed data. Readers can directly access the website through the link http://meni.singlecell.info:3000/, and perform online analyses of the results, comparison of differential genes between two special clusters, or data downloading.

In the design of this study, we only partitioned samples into two health states: normal and degenerated. If we distinguish the disease severity of the meniscus samples, we could establish more fine-grained links between the molecular profiles and the disease. Another limitation of this study is that females make up the majority of our donors (*Supplementary file 1*), since OA is more common in middle-aged and older females. The cellular profiles observed in this study may better represent the female population. We observed a small number of nerve tissue cells, which also hindered detailed analysis of them. Our functional coordination of chondrocyte subtypes was confined to the meniscus environment, even though there were signs that chondrocytes across different cartilage tissues were not distinct and had many common features. We hope a future molecular coordination framework could integrate chondrocytes across multiple cartilage tissues, different health states, anatomical regions, and differentiation stages, guiding the precise taxonomy of chondrocytes when more single-cell cartilage studies are released.

In conclusion, we systematically profiled the cellular diversities in the inner and outer meniscus and reported the microenvironmental alterations in the healthy and degenerated states. Our study is an informative complement to the existing meniscal single-cell sequencing data and provides an important reference for the study of meniscal degeneration. The study suggested that we should pay attention to the blood vessel's functional duality in acute and chronic inflammation. Preserving the pericyte-like cells wrapping around the vessels might be a potential strategy to alleviate chronic degeneration. Systematically coordinating of angiogenesis, inflammation response, ECM catabolism, and preventing their mutual reinforcement could be a strategy for delaying or reversing meniscus

degeneration. The meniscus is an important and representative fibrocartilage in the human body. We speculated that the cellular and molecular mechanisms observed in this study on meniscal degeneration, such as angiogenesis and inflammation, could also apply to intervertebral discs and articular cartilage degeneration. It could shed new light on the diagnosis and treatment of other degeneration-associated musculoskeletal diseases like low back pain and joint pain.

## Acknowledgements

The authors are grateful to Dr. Tao Wu from the Institute of Immunology, Tsinghua University, for his helpful discussions. The authors are grateful to Kaibo Zhang, Zhong Zhang, Beini Mao, Sike Lai, and Menglin Yao for the help of cell isolations and sample preparations. The authors are grateful to Yi Zhang, Yue Li, and Wanli Zhang for the help of multiplex immunofluorescence staining.

The experiments were approved by Ethics Committee on Biomedical Research, West China Hospital of Sichuan University with an identifier of No. 2020-921. All individuals have been informed of the experiments and analyses, and they have given consent to use their tissue samples for this research and publication.

## Additional information

### Funding

| Funder | Grant reference number | Author |
| --- | --- | --- |
| National Natural Science Foundation of China | 81972123 | Weili Fu |
| National Natural Science Foundation of China | 82172508 | Weili Fu |

The funders had no role in study design, data collection, and interpretation, or the decision to submit the work for publication.

### Author contributions

Weili Fu, Conceptualization, Funding acquisition, Writing – original draft, Project administration, Writing – review and editing; Sijie Chen, Conceptualization, Resources, Data curation, Software, Funding acquisition, Visualization, Methodology, Writing – original draft; Runze Yang, Formal analysis, Validation, Writing – original draft; Chen Li, Software, Visualization, Methodology; Haoxiang Gao, Software; Jian Li, Conceptualization, Writing – review and editing; Xuegong Zhang, Conceptualization, Data curation, Supervision, Funding acquisition, Project administration, Writing – review and editing

### Author ORCIDs
Weili Fu  http://orcid.org/0000-0003-4438-2760

### Ethics
The experiments were approved by Ethics Committee on Biomedical Research, West China Hospital of Sichuan University with an identifier of No. 2020-921. All individuals have been informed of the experiments and analyses, and they have given consent to use their tissue samples for this research and publication.

### Decision letter and Author response
Decision letter https://doi.org/10.7554/eLife.79585.sa1
Author response https://doi.org/10.7554/eLife.79585.sa2

## Additional files

### Supplementary files
• Supplementary file 1. Donor and sample information. A table contains the descriptions of sample

names, cell counts, cell viabilities, sample volumes, estimated numbers of cells loaded, donor states, donor genders, anatomical locations, estimated numbers of cells after filtering, body mass index (BMI) of donor, and ages of donor.

• Supplementary file 2. Differentially expressed genes of each cluster. A list of differentially expressed genes of each cluster. The table contains gene symbols, corresponding clusters, average log2 fold-change values, percents of cells expressing this gene in groups 1 and 2, p values, and adjusted p-values.

• MDAR checklist

## Data availability

Data are available in a public, open access repository. The single-cell RNA-seq data, cluster annotations are available at GSA for human (https://ngdc.cncb.ac.cn/gsa-human/) with the accession number PRJCA008120.

The following dataset was generated:

| Author(s) | Year | Dataset title | Dataset URL | Database and Identifier |
|---|---|---|---|---|
| Weili Fu1, Sijie Chen, Runze Yang, Chen Li, Haoxiang Gao, Jian Li, Xuegong Zhang | 2022 | Single-cell transcriptomic profiling of healthy and degenerated human menisci | https://ngdc.cncb. ac.cn/bioproject/ browse/PRJCA008120 | BioProject, PRJCA008120 |

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
