## [Editor Report]

This paper will be of interest to researchers studying meniscus homeostasis and knee osteoarthritis. It uncovers distinct subtypes of cell populations in the inner and outer parts of the human meniscus using single-cell RNA sequencing. In particular, this work further identifies how alterations in meniscal cell populations may contribute to inflammation and osteoarthritis and thus serves as a resource paper for the field.

---

## [Decision Letter]

**Decision letter after peer review:**

Thank you for submitting your article "Depiction of cell atlas and analysis of microenvironment alterations in human meniscus degeneration using single-cell transcriptomic" for consideration by *eLife*. Your article has been reviewed by 3 peer reviewers, one of whom is a member of our Board of Reviewing Editors, and the evaluation has been overseen by Kathryn Cheah as the Senior Editor. The following individual involved in review of your submission has agreed to reveal their identity: Chia-Lung Wu (Reviewer #2).

Essential revisions:

1. There are many definitive statements about inferred results. The authors jump to too many mechanistic conclusions only based cell profiling. For example, "The functional analysis identified subtypes that contributed to the angiogenesis/anti-angiogenesis process, and subtypes that contributed to the ECM construction/disassembly process." The result is not the issue, but rather the way it is worded. The authors have not directly tested that these subtypes contributed to the ECM construction, rather this work can only suggest that this subtype of cells contributed to ECM construction by the genes observed to be expressed. This is a very descriptive paper and the authors must consistently and carefully revise this paper to reflect that it is being published as resource paper, not a research paper with definitive conclusions. Many claims in the paper are not backed up by data.

2. Page 5, This is a resource paper. Therefore, the authors must tone down the statements for Sun et al.' study. Indeed. The current work may still be co-founded by limited sample size. Eight patients is still a small sample size compared to a true clinical study or even other scRNA-seq studies such as in the field of rheumatoid arthritis. It is suggested that the authors emphasize the strengths of their data (i.e., inner and outer zone) sequencing and mention this present study is actually a complementary work to Sun et al.'s work.

3. Another example is "2.7. Endothelial cells, immune cells and chondrocytes form a positive feedback loop of degeneration" Given the data available this was not tested. A feedback loop might be inferable, but it was not proven.

4. The same issues apply to the cell-cell interaction analysis. These results are inferred from gene expression, but active interaction with a measured response from that reaction was not actively examined. In total, the authors need to carefully reword this paper to avoid implying experiments that just were not done.

5. Please note that the figures and supplemental figures are not introduced/explained in order in the current format. This makes the manuscript very challenging to read as the readers need to come back and forth to find the figures that the authors are discussing by flipping through several pages. Please consider reordering the figures or making a better flow of the manuscript.

6. The major novelty and strength of this work is that the samples were collected and sequenced from inner and outer zone of the human meniscus, respectively. As the authors indicated that different zonal areas of a meniscus have varied recovery capabilities in response to injury. However, it remains unknown whether inner or outer zone of a meniscus is the primary contributor for joint inflammation in osteoarthritis (OA). I believe that the authors should take advantage of their current datasets and provide insights into this unanswered question. For example, if chondrocyte 2 (Ch2) and PRG4-enriched Ch3 chondrocyte populations are main population observed in degenerated meniscus as the authors stated, in what zones of a meniscus of these cells were detected? And it might provide further impact in the field if the authors could elaborate on the possible mechanisms leading to the increased Ch2 and Ch3 in the degenerated meniscus (i.e., are Ch2 and Ch3 induced into proliferating status or other chondrocyte population differentiate into Ch2/Ch3 due to inflammation?)

7. In the current analysis, 4 major seq data were integrated together: inner and outer zone seq data as well as normal and degenerated meniscus seq data. This integration was also mostly used for interpretation. Although this approach helped identify conserved subset populations (in particular chondrocytes) across inner and outer zone, it may also overlook rare or small populations that exist only in a given zonal area. It appeared that in Figure 1C-G were the integration from normal and diseased tissues for inner and outer zone seq data, respectively. Have the authors re-clustered ACTA2+ cells in the inner and outer part separately and see if they can identify DegP markers and possible topological connection between FCP and DegP? Also, please change the Figure legends of Figure 1C-F so it is easier for readers to understand these data were integrated from both normal and degenerated menisci.

8. As OA is sex-dependent disease, do the authors notice the cell populations changes in the inner and outer zone of meniscus from male and female donors?

9. The authors attempted to compare their cell population with the ones identified by Sun et al. (Supplemental figure S9-S11). The figure resolution is too low, particular for Supplemental figure S9C. Furthermore, using hierarchical clustering and heatmaps with complete transcriptome of each cell population from both studies should be a better way to perform a comprehensive comparison rather than only rely on a few markers. Such a comparison is essential to link two studies together.

10. In the introduction, the authors mentioned "The cellular and molecular changes during meniscal degeneration we observed may also be the mechanism of a variety of other cartilage degenerations." sine meniscus composes mainly fibrochondrocytes while articular cartilage is a hyaline cartilage. The reviewers are not quite sure both tissues have exactly the same mechanisms underlying matrix degeneration. Please either rephrase this statement or provide strong evidence to support this claim.

11. The introduction includes justification for the study, but some sentences have logical flaws which make it difficult to read. For example, Page 4, paragraph 2, sentence 1: Cell landscape for better understanding. But the reason for this cannot be BECAUSE content of ECM is produced by meniscal cells.

12. A heatmap with markers for each identified cell populations was used to distinguish unique cell subset (Figure 1F). However, gene annotation of the heatmap does not provide exact location of a given gene. Violin plots may be better method to demonstrate this. Please refer to Figure 4F in Sun et al.'s study. This violin may also replace Figure 2G it is not certain that Figure 2G provides clear and particular useful information to the readers and Figure 2G may be placed in the supplemental materials.

13. Figure 1E/H are difficult to read. Please directly indicate which donors belong to normal and which donors belong to degenerated conditions on the figure.

14. Please provide relevant references demonstrating pericytes expressing ACTA2 and how pericytes can be distinguished from smooth muscle cells.

15. For GRN analysis (Supplemental figure S8B), please indicate which part of GRN belong to specific chondrocyte populations and their associated modules. And what part of the GRN are shared by different chondrocyte populations? Indeed, the authors may use the same color to indicate shared genes (nodes), which can make this GRN easier to read and understand.

16. For GRN analysis, as this is an analysis of co-regulated genes/modules, please elaborate how the direction between the nodes is determined in the GRN? Mainly based on previous reported literature? If this is the case, this could be one limitation of this GRN as not all TFs and their possible downstream targets are validated in the context of chondrocytes.

17. The authors mentioned that "Antiangiogenic genes like COL4A1, COL4A2, COL18A1 are highly expressed in PCL.1 (Supplementary Figure S12B)." Please provide relevant references showing these genes are anti-angiogenic genes. Furthermore, it appears that COL4A1 is up-regulated in PCL2 rather than PCL1 Will violin plots provide better visualization? Are these genes in fact DEGs between PCL1 and PCL2 (i.e., adjusted p value <0.0.5)?

18. Page 11, the authors mentioned "We inferred pericyte-like cells in meniscal microenvironments played similar anti-angiogenesis roles". Please provide relevant evidence or previous findings showing pericytes have an anti-angiogenesis role.

19. Page 13, the authors mentioned "Matrix remodeling-associated protein 5 (MXRA5) is a TGF-beta1 regulated protein". Please provide relevant evidence or previous findings.

20. Please indicate the zones (inner or outer) of meniscus the IF was performed (Figure 4A-H).

21. Please perform statistical analysis on Figure 5D.

22. Page 16-17, references are required to support the authors claim regarding surface markers of different subsets of endothelia cells. Page 17, references are required for ACKR1 function.

23. Figure 5J, cell-cell crosstalk of CXCL8 but not SERPINE1-PLAUR was explained. Are there any key messages the authors would like to deliver for SERPINE1-PLAUR signaling?

24. DAMP signaling is mentioned in page 18 as "Damages in meniscal tissues broke the ECM and released damage-associated molecular patterns (DAMP) that initiated inflammatory responses" References are required for this. Indeed, cartilage matrix degradation could also activate DAMP signaling and initiated the inflammation. It remains unclear whether cartilage or meniscal tissues are degraded first in the development of OA and this may be highly like to be injury-dependent.

25. Page 22, the authors mentioned "While the pericyte stabilizes the blood vessel in health, the detachment and decrease of the pericyte may lead to pathological angiogenesis". Relevant references are required to support this statement.

26. The experimental design is not consistent between the table in the supplemental material (6+6 menisci) and methods section (4+4 menisci).

27. It is not clear how the degeneration of menisci was defined.

28. Page 22, paragraph 1: How many cells were loaded on chip? Standardized or simply all from isolation?

---

## [Author Response]

Essential revisions:1. There are many definitive statements about inferred results. The authors jump to too many mechanistic conclusions only based cell profiling. For example, "The functional analysis identified subtypes that contributed to the angiogenesis/anti-angiogenesis process, and subtypes that contributed to the ECM construction/disassembly process." The result is not the issue, but rather the way it is worded. The authors have not directly tested that these subtypes contributed to the ECM construction, rather this work can only suggest that this subtype of cells contributed to ECM construction by the genes observed to be expressed. This is a very descriptive paper and the authors must consistently and carefully revise this paper to reflect that it is being published as resource paper, not a research paper with definitive conclusions. Many claims in the paper are not backed up by data.

Thanks for the comments and suggestions. We followed them and revised the wordings of the observations and hypothetical discussions carefully throughout the paper.

2. Page 5, This is a resource paper. Therefore, the authors must tone down the statements for Sun et al.' study. Indeed. The current work may still be co-founded by limited sample size. Eight patients is still a small sample size compared to a true clinical study or even other scRNA-seq studies such as in the field of rheumatoid arthritis. It is suggested that the authors emphasize the strengths of their data (i.e., inner and outer zone) sequencing and mention this present study is actually a complementary work to Sun et al.'s work.

Thank you for your suggestions. We accepted them and have made corresponding revisions in the Introduction section accordingly (page 4).

3. Another example is "2.7. Endothelial cells, immune cells and chondrocytes form a positive feedback loop of degeneration" Given the data available this was not tested. A feedback loop might be inferable, but it was not proven.

We agreed that we had only done computational experiments to support the "positive feedback loop" statement. Now we have tuned down the discussion to a more rigorous one.

4. The same issues apply to the cell-cell interaction analysis. These results are inferred from gene expression, but active interaction with a measured response from that reaction was not actively examined. In total, the authors need to carefully reword this paper to avoid implying experiments that just were not done.

We have reworded the cell-cell interaction analysis part and stated that the interactions were not examined by experiments.

5. Please note that the figures and supplemental figures are not introduced/explained in order in the current format. This makes the manuscript very challenging to read as the readers need to come back and forth to find the figures that the authors are discussing by flipping through several pages. Please consider reordering the figures or making a better flow of the manuscript.

Sorry for that. We have re-arranged the (supplemental) figures in the revision. Here are the relationships between the old and new figures:

Original Supplementary Figure S1 Figure 1 —figure supplement 1

Original Supplementary Figure S2 Figure 1 —figure supplement 2

Original Supplementary Figure S3 Figure 2 —figure supplement 1

Original Supplementary Figure S4 Figure 2 —figure supplement 2

Original Supplementary Figure S5 Figure 3 —figure supplement 4

Original Supplementary Figure S6 Figure 3 —figure supplement 5

Original Supplementary Figure S7 Figure 3 —figure supplement 6

Original Supplementary Figure S8 Figure 2 —figure supplement 3

Original Supplementary Figure S9 Figure 3 —figure supplement 2

Original Supplementary Figure S10 Figure 2 —figure supplement 4

Original Supplementary Figure S11 Figure 2 —figure supplement 4

Original Supplementary Figure S12 Figure 3 —figure supplement 1

Original Supplementary Figure S13 Figure 5 —figure supplement 2

New "Figure 3 —figure supplement 3" inserted

New "Figure 3 —figure supplement 7" inserted

Main figures keep their previous orders.

6. The major novelty and strength of this work is that the samples were collected and sequenced from inner and outer zone of the human meniscus, respectively. As the authors indicated that different zonal areas of a meniscus have varied recovery capabilities in response to injury. However, it remains unknown whether inner or outer zone of a meniscus is the primary contributor for joint inflammation in osteoarthritis (OA). I believe that the authors should take advantage of their current datasets and provide insights into this unanswered question. For example, if chondrocyte 2 (Ch2) and PRG4-enriched Ch3 chondrocyte populations are main population observed in degenerated meniscus as the authors stated, in what zones of a meniscus of these cells were detected? And it might provide further impact in the field if the authors could elaborate on the possible mechanisms leading to the increased Ch2 and Ch3 in the degenerated meniscus (i.e., are Ch2 and Ch3 induced into proliferating status or other chondrocyte population differentiate into Ch2/Ch3 due to inflammation?)

Thank you for your great questions. We are also curious to these inner/outer difference questions. We performed some new experiments and inferred outer menisci could contributed more to the joint inflammation in osteoarthritis. We also found that the proliferative population Ch.5 could mainly differentiate into Ch.2 according to our mapping analysis.

We added texts discussing the zonation differences and primary contributor of inflammation at the end of section 2.5, and added texts discussing Ch.5 might give rise to Ch.2 in section 2.4 Ch.5 descriptions.

7. In the current analysis, 4 major seq data were integrated together: inner and outer zone seq data as well as normal and degenerated meniscus seq data. This integration was also mostly used for interpretation. Although this approach helped identify conserved subset populations (in particular chondrocytes) across inner and outer zone, it may also overlook rare or small populations that exist only in a given zonal area. It appeared that in Figure 1C-G were the integration from normal and diseased tissues for inner and outer zone seq data, respectively. Have the authors re-clustered ACTA2+ cells in the inner and outer part separately and see if they can identify DegP markers and possible topological connection between FCP and DegP? Also, please change the Figure legends of Figure 1C-F so it is easier for readers to understand these data were integrated from both normal and degenerated menisci.

Thank you for your suggestions. It's true that over-integration often overlooks rare or small populations. We have now re-clustered the ACTA2+ cells and the chondrocytes in the inner/outer part separately as well as in the normal/degeneration states separately. We did find rare populations after re-clustering of the subset but the rare populations could be doublets arisen from technical noise rather than true biological signal. We did not find significant DegP marker expressions either.

We performed a series of integration experiments to check the over-integration issues and to find the identities of the new clusters. Since integration strength issues is out of the scope of our paper and numerous new figures were generated, we did not include these figures into our supplementary materials. We have put the figures at FigShare 10.6084/m9.figshare.21432552 (https://figshare.com/articles/figure/Single-cell_analysis_reveals_cell_heterogeneity_and_microenvironment_alterations_in_human_meniscal_degeneration/21432552) so that you can check the details if necessary.

We also changed the Figure legends of Figure 1C-F.

8. As OA is sex-dependent disease, do the authors notice the cell populations changes in the inner and outer zone of meniscus from male and female donors?

Since OA has a higher prevalence in women than men, we are interested in this too. However, in our collected samples, 6 patients (degenerated: A _inner/outer_, B _inner/outer_, C1 _inner_, D2 _outer_, normal: E _inner/outer_, F _inner/outer_) are all female, 2 patients (normal: G11 _outer_, H12 _inner_) are male and unpaired. We can't draw solid conclusions with the limited sample sizes.

9. The authors attempted to compare their cell population with the ones identified by Sun et al. (Supplemental figure S9-S11). The figure resolution is too low, particular for Supplemental figure S9C. Furthermore, using hierarchical clustering and heatmaps with complete transcriptome of each cell population from both studies should be a better way to perform a comprehensive comparison rather than only rely on a few markers. Such a comparison is essential to link two studies together.

Sorry for the inconvenience. The original supplemental figure S9-S11 contains many scatter plots, each of which contains ~45k data points. Hence the file volumes of the vector graphs were so large that we had to rasterized and squeezed them to fulfill the uploading system requirements. We have now switched to a better way to present expressional values on these dense overlaying points – kernel density estimation plots of the expressions provided by R package Nebulosa (Alquicira-Hernandez and Powell, *Bioinformatics*, 2021, doi: 10.1093/bioinformatics/btab003). The new plots present smoothed expressions that can better characterize the distribution of the values without point-crowding flaws. We updated the feature plots in the original Supplemental Figure S9 and placed them in the new "Figure 2 —figure supplement 4", then updated the original S10, S11 plots and placed them in the new "Figure 3 —figure supplement 2".

As for the comparisons between our data and the public data, we have now performed a comprehensive integration provided by Seurat algorithm (Stuart, Butler, Hoffman, et al. *Cell*, 2019. doi: 10.1016/j.cell.2019.05.031). See detailed descriptions in at the end of section 2.4.

10. In the introduction, the authors mentioned "The cellular and molecular changes during meniscal degeneration we observed may also be the mechanism of a variety of other cartilage degenerations." sine meniscus composes mainly fibrochondrocytes while articular cartilage is a hyaline cartilage. The reviewers are not quite sure both tissues have exactly the same mechanisms underlying matrix degeneration. Please either rephrase this statement or provide strong evidence to support this claim.

Thank you for your suggestions. We rephrased the statement. The cartilage-like tissues did share many common profiles. We now cited a review article comparing the matrix microenvironment of meniscus, articular cartilage and nucleus pulposus. (Chen, Fu, Wu, et al. Cell Tissue Res, 2017. https://doi.org/10.1007/s00441-017-2613-0)

11. The introduction includes justification for the study, but some sentences have logical flaws which make it difficult to read. For example, Page 4, paragraph 2, sentence 1: Cell landscape for better understanding. But the reason for this cannot be BECAUSE content of ECM is produced by meniscal cells.

Thanks. We have updated the texts.

12. A heatmap with markers for each identified cell populations was used to distinguish unique cell subset (Figure 1F). However, gene annotation of the heatmap does not provide exact location of a given gene. Violin plots may be better method to demonstrate this. Please refer to Figure 4F in Sun et al.'s study. This violin may also replace Figure 2G it is not certain that Figure 2G provides clear and particular useful information to the readers and Figure 2G may be placed in the supplemental materials.

Thank you for your suggestions. We guess you meant Figure 2F. In Figure 2F, we visualized an overall expression profile of chondrocytes and PCLs using a heatmap with selected markers. We referred to the Fig4F in Sun et al.'s study and made new stacked violin plots in Figure 2 —figure supplement 1C. We also replaced the GSVA score plots in Figure 2G with new stacked violin plots and put the old radar plots in Figure 2 —figure supplement 2H.

We have now provided a full list of gene annotations in the Fig2F top-to-bottom order in Supplementary File 2. We kept the current layout of Fig2F because we believe it is still an informative figure conveying rich information. With the heatmap, we could easily add status/anatomy annotation bars on the top of the figure besides the cell type group information. The rows in the heatmap are top differentially expressed genes (DEGs) in each group. These DEGs captured the major source of transcriptional variations well and reveals the distinct boundaries of clusters. We did not show all the DEGs (and they have been listed in Supplementary File 2 now) for two reasons: (1) Some DEGs have very high or specific expressions in a group or have been reported in earlier literatures; (2) The space of the figure did not allow for all rows annotated with gene symbols.

13. Figure 1E/H are difficult to read. Please directly indicate which donors belong to normal and which donors belong to degenerated conditions on the figure.

Thanks. we have revised the Figure 1E/H and explicitly annotated the composition of the samples in the normal and degenerated groups in the figure.

14. Please provide relevant references demonstrating pericytes expressing ACTA2 and how pericytes can be distinguished from smooth muscle cells.

Thank you for the suggestion. We have added several literature references and some descriptions demonstrating pericytes expressing ACTA2 at the beginning of the Section 2.3

We have to admit that it is hard to distinguish pericyte from smooth muscle cells purely based on transcriptomes. That is why we used the term "pericyte-like" instead of "pericyte". Pericytes (PCs) and smooth muscle cells (SMCs) originate from the same mesenchymal lineage, share many marker expressions like PDGFRB, αSMA, CD13, CD146, and both can serve as mural structure around the blood vessels structures. We have now added explanations in Section 2.3.

15. For GRN analysis (Supplemental figure S8B), please indicate which part of GRN belong to specific chondrocyte populations and their associated modules. And what part of the GRN are shared by different chondrocyte populations? Indeed, the authors may use the same color to indicate shared genes (nodes), which can make this GRN easier to read and understand.

Thank you. We have now updated the network in Figure 2 —figure supplement 3B (the original supplementary figure S8) to a refined version and colored the network modules with its corresponding cell subtypes. Color shaded polygons highlighted GRN cliques with condensed nodes and links. Detailed GRN descriptions can be found in at the end of section 2.2

16. For GRN analysis, as this is an analysis of co-regulated genes/modules, please elaborate how the direction between the nodes is determined in the GRN? Mainly based on previous reported literature? If this is the case, this could be one limitation of this GRN as not all TFs and their possible downstream targets are validated in the context of chondrocytes.

We used a popular GRN analysis tool SCENIC to derive the modules (Aibar, S., *et al. Nat Methods* 2017). SCENIC combines in-house data GRN predictions with database records, so the GRN we identified is not limited to the previous reported sets. We provided detailed descriptions of the GRN algorithm in Methods 4.7.

17. The authors mentioned that "Antiangiogenic genes like COL4A1, COL4A2, COL18A1 are highly expressed in PCL.1 (Supplementary Figure S12B)." Please provide relevant references showing these genes are anti-angiogenic genes. Furthermore, it appears that COL4A1 is up-regulated in PCL2 rather than PCL1 Will violin plots provide better visualization? Are these genes in fact DEGs between PCL1 and PCL2 (i.e., adjusted p value <0.0.5)?

Thanks for your suggestions. The collagens encoded by these genes could be further converted to anti-angiogenic factors (arresten, canstatin, endostatin). We have clarified the genes' characteristics with some new statements and added the references (de Castro Bras and Frangogiannis, 2020; Marneros and Olsen, 2001) in Section 2.3.

We double checked Figure 2E and confirmed that COL4A1 should be up-regulated in PCL.1 rather than PCL.2. To make this clear, we made violin plots with Wilcoxon test p-values to better visualize the variations (Figure 3 —figure supplement 1 E). Among these genes, COL4A1, COL4A2 are DEGs and upregulated in PCL.1 (adjusted p-value<0.05).

18. Page 11, the authors mentioned "We inferred pericyte-like cells in meniscal microenvironments played similar anti-angiogenesis roles". Please provide relevant evidence or previous findings showing pericytes have an anti-angiogenesis role.

Thank you for your suggestions. Previous studies have shown pericyte have pro-angiogenesis characteristics. What we hope to convey is that pericyte-like cells up-regulated several genes that may have anti-angiogenesis functions, e.g., COL4A1, COL4A2, COL18A1, whose product could be further converted into angiogenesis inhibitors like arresten, canstatin, endostatin.

We have revised the statement involving PCL functional analysis in Section 2.3 and used descriptive words to introduce these upregulated genes.

19. Page 13, the authors mentioned "Matrix remodeling-associated protein 5 (MXRA5) is a TGF-beta1 regulated protein". Please provide relevant evidence or previous findings.

We cited an article to support this statement: Poveda J, Sanz AB, Fernandez-Fernandez B, *et al.* MXRA5 is a TGF-β1-regulated human protein with anti-inflammatory and anti-fibrotic properties. *J Cell Mol Med*. 2017;21(1):154-164. doi:10.1111/jcmm.12953

20. Please indicate the zones (inner or outer) of meniscus the IF was performed (Figure 4A-H).

As Reviewer suggested that it is indeed better to indicate the inner and outer zones of meniscus in Figure 4A-H for clearly identifying the structure of the meniscus. We have explicitly marked the inner and outer areas of meniscus in the figures.

21. Please perform statistical analysis on Figure 5D.

As requested by the reviewer, we have performed statistical analysis, calculated the p-value and added them to the legends of Figure 5D.

22. Page 16-17, references are required to support the authors claim regarding surface markers of different subsets of endothelia cells. Page 17, references are required for ACKR1 function.

Done.

23. Figure 5J, cell-cell crosstalk of CXCL8 but not SERPINE1-PLAUR was explained. Are there any key messages the authors would like to deliver for SERPINE1-PLAUR signaling?

SERPINE1-PLAUR crosstalk also have a chemotaxis function, so we showed it together with CXCL8. We take CXCL8 as an example because it is more common in the figure. We did not intent to deliver key messages. As a resource paper, we predicted a lot of cell types crosstalks upon readers' queries. We listed a few in the main figure, and there are a lot in Figure 5 —figure supplement 2.

24. DAMP signaling is mentioned in page 18 as "Damages in meniscal tissues broke the ECM and released damage-associated molecular patterns (DAMP) that initiated inflammatory responses" References are required for this. Indeed, cartilage matrix degradation could also activate DAMP signaling and initiated the inflammation. It remains unclear whether cartilage or meniscal tissues are degraded first in the development of OA and this may be highly like to be injury-dependent.

According to your suggestions, we added two references to support this statement:

1. Martel-Pelletier, J., Barr, A. J., Cicuttini, F. M., Conaghan, P. G., Cooper, C., Goldring, M. B., Goldring, S. R., Jones, G., Teichtahl, A. J. and Pelletier, J. P. 2016. Osteoarthritis. *Nat Rev Dis Primers*, 2, 16072.

2. Chen, D., Shen, J., Zhao, W., Wang, T., Han, L., Hamilton, J. L. and Im, H. J. 2017. Osteoarthritis: toward a comprehensive understanding of pathological mechanism. *Bone Res,* 5**,** 16044.

Both cartilage and meniscal tissue injury produces DAMP signals. We are also curious which tissue degraded first in the development of OA. However, we can't answer this question using current single-cell data. We plan to use more clinical data combined with dedicated causal inference designs to explore this problem.

25. Page 22, the authors mentioned "While the pericyte stabilizes the blood vessel in health, the detachment and decrease of the pericyte may lead to pathological angiogenesis". Relevant references are required to support this statement.

The statement was inaccurate. We changed it to "While the pericyte stabilizes the blood vessel in health, the detachment and decrease of the pericyte may lead to pathological states" and added several references (Figueiredo et al., 2020; Armulik et al., 2011; Dieguez-Hurtado et al., 2019).

26. The experimental design is not consistent between the table in the supplemental material (6+6 menisci) and methods section (4+4 menisci).

We have 6 degenerated samples from 4 patients, two patients of which contributed the paired samples (2+2), another two contributed unpaired samples (1+1). We also have 6 normal samples from 4 donors, two paired (2+2) and two unpaired (1+1). We further clarified sample-donor texts in the method section.

27. It is not clear how the degeneration of menisci was defined.

Thank you for your reminding. We added the definition of degeneration of meniscus in the introduction. Degeneration of menisci is a slow-developing disease that usually characterizes a horizontal cleavage of the meniscus in middle-aged or older people. Patients usually have no clear history of acute knee injury. Such meniscus lesions are common in the general population and often found incidentally through knee MRI. (https://pubmed.ncbi.nlm.nih.gov/29114633/).

28. Page 22, paragraph 1: How many cells were loaded on chip? Standardized or simply all from isolation?

Cells were loaded on chip without fluorescence-activated cell sorting. We provided the number of cells loaded in Supplementary File 1.